# RG flows in de Sitter: c-functions and sum rules

**Manuel Loparco**

*Fields and Strings Laboratory, Institute of Physics*
*École Polytechnique Fédéral de Lausanne (EPFL)*
*Route de la Sorge, CH-1015 Lausanne, Switzerland*

*E-mail:* manuel.loparco@epfl.ch

ABSTRACT: We study the renormalization group flow of unitary Quantum Field Theories on two-dimensional de Sitter spacetime and on the Euclidean two-sphere of radius $R$. We prove the existence of two functions $c_1(R)$ and $c_2(R)$ which interpolate between the central charges of the UV and of the IR fixed points of the flow when tuning the radius $R$ while keeping the mass scales of the theory fixed. $c_1(R)$ is constructed from certain components of the two-point function of the stress tensor evaluated at antipodal separation. $c_2(R)$ is the spectral weight of the stress tensor over the $\Delta = 2$ discrete series. This last fact implies that the stress tensor of any unitary QFT in $S^2/\mathrm{dS}_2$ must interpolate between the vacuum and states in the $\Delta = 2$ discrete series irrep. We verify that the c-functions are monotonic for intermediate radii in the free massive boson and free massive fermion theories, but we lack a general proof of said monotonicity. We derive a variety of sum rules which relate the central charges and the c-functions to integrals of the two-point function of the trace of the stress tensor and to integrals of its spectral densities. The positivity of these formulas implies $c^{\mathrm{UV}} \geq c^{\mathrm{IR}}$. In the infinite radius limit the sum rules reduce to the well known formulas in flat space. Throughout the paper, we prove some general properties of the spectral decomposition of the stress tensor in $S^{d+1}/\mathrm{dS}_{d+1}$.

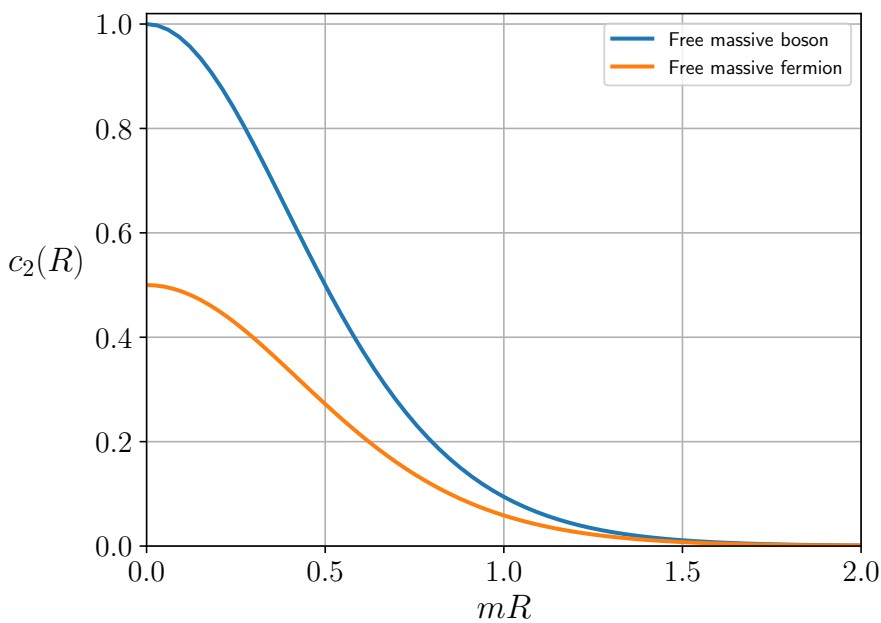

**Figure 1:** $c_2(R)$ for the free massive fermion and free massive boson flows in $\mathrm{dS}_2$.

# 1 Introduction

Unitary and Lorentz invariant quantum field theories (QFTs) in two dimensions describe renormalization group (RG) flows between two conformal field theories (CFT), one in the long distance (IR) regime, and one in the short distance (UV) regime. Zamolodchikov's seminal paper [1] showed that to each flow one can assign a function which is monotonic in the scales of the theory, and which asymptotes to the central charges of the two CFTs at the fixed points. The difference between the two central charges $\Delta c \equiv c^{\mathrm{UV}} - c^{\mathrm{IR}}$ is positive, a fact that is usually referred to as the c-theorem, and it can be related to sum rules involving integrals of observables computed along the flow [2–5]

$$\Delta c = 6\pi^2 \int_0^\infty r^3 dr \langle \Theta(r)\Theta(0)\rangle = 12\pi \int_0^\infty \frac{ds}{s^2} \varrho_\Theta(s) \,, \tag{1.1}$$

where $r$ is a radial coordinate on the Euclidean plane, $\Theta$ is the trace of the stress tensor and $\varrho_\Theta$ is its spectral density over the $s = m^2 > 0$ unitary irreducible representations (UIRs) of the Poincaré group in two dimensions.

The existence of a function that is monotonic under RG flows implies that the flows themselves are irreversible, giving a quantitative basis to the intuition that there is a loss of degrees of freedom when "zooming out" and coarse graining in QFT. It is thus interesting to establish the existence of other RG-monotonic functions (also called c-functions) for QFTs in higher dimensions and on curved backgrounds, providing new general constraints on RG flows.

In [6], Cardy conjectured that the one-point function of $\Theta$ integrated over a sphere could be a c-function in spacetimes with an even number of dimensions. This fact was proven in 4d by Komargodski and Schwimmer [7] and is called the $a$-theorem, since said integral isolates the coefficient of the Euler density in the trace anomaly of the UV and IR CFTs, usually denoted as $a$. In 3d, Casini and Huerta proved the $F$-theorem [8], stating that the finite part of the free energy on a three-sphere satisfies $F^{\mathrm{UV}} \geq F^{\mathrm{IR}}$. This had been conjectured in [9], and in [10–12] it was proposed that $\sin\left(\frac{\pi}{2}d\right)\log Z_{S^d}$ with $Z_{S^d}$ being the partition function of the theory on $S^d$, could be the generalization of $F$ to any dimension. While many checks and no counter examples are known, there is still no proof for this last statement.

In this work, we focus on RG flows in reflection positive QFTs on a two-dimensional Euclidean sphere $S^2$, or equivalently unitary QFTs in two-dimensional de Sitter spacetime $\mathrm{dS}_2$. The study of RG flows in dS has a long history, see for example [13–43]. Leveraging recent advances in understanding non-perturbative unitarity [44–49] and analyticity [50–54], our main result is to prove the existence of two functions $c_1(R)$ and $c_2(R)$ which interpolate between the central charges at the fixed points of the RG flow as we tune the radius of $S^2/\mathrm{dS}_2$ while keeping the mass scales of the theory fixed. At infinite radius we recover $c^{\mathrm{IR}}$ and at vanishing radius $c^{\mathrm{UV}}$. In contrast to the $F$-theorem and its generalization, $c_1(R)$ and $c_2(R)$ are related to correlation functions of a local operator, namely the stress tensor. In the examples of a free massive boson and free massive fermion, we find that these functions are also monotonic for intermediate $R$, although we do not have

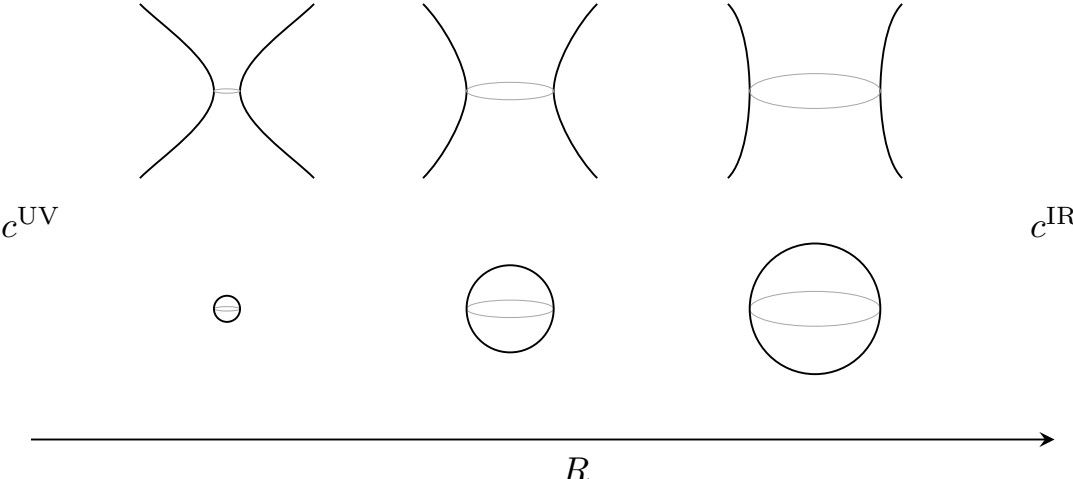

**Figure 2:** Both in Euclidean and Lorentzian signature the functions $c_1$ and $c_2$ interpolate between the central charges of the CFTs at the endpoints of RG flows.

a theory-independent proof of this fact. We further consider the example of the massless Schwinger model, in which $c_1$ and $c_2$ match the same functions as in the free massive boson theory, hinting towards the fact that there exists a field redefinition that relates the two theories in dS, just as in flat space.

The general point we advocate for in this work is that the radius of the sphere provides a valuable IR regulator which can be used to follow RG flows in any QFT of interest.

**Outline** The paper is structured as follows. In section 2, we define $c_1(R)$ (2.29) and derive a sum rule (2.30) to compute $c^{\mathrm{UV}} - c_1(R)$ in terms of an integral of the two-point function of the trace of the stress tensor over the chordal distance. We take a flat space limit and recover Cardy's sum rule [2], showing that $c_1(\infty) = c^{\mathrm{IR}}$. Then, we use the Källén-Lehmann decomposition in de Sitter [44–46, 55] to prove a sum rule for $c^{\mathrm{UV}} - c_1(R)$ in terms of the spectral densities of the trace of the stress tensor (2.39). Its flat space limit reproduces the sum rule from [3]. Finally, we show that $c_1(0) = c^{\mathrm{UV}}$.

In section 3, we show that conservation greatly simplifies the spectral decomposition of the stress tensor (3.6) in any number of dimensions. In two dimensions, we show that there are only three independent spectral densities: one for the principal series, one for the complementary series and one for the $\Delta = 2$ discrete series (3.8). The latter quantity is precisely $c_2(R)$. We show that $c_2(\infty) = c^{\mathrm{IR}}$ and $c_2(0) = c^{\mathrm{UV}}$, and we prove sum rules for $c^{\mathrm{UV}} - c_2(R)$ in terms of integrals of the other spectral densities (3.15) and in terms of an integral of the two-point function of the trace of the stress tensor in position space (3.19). We also show sum rules which compute $c^{\mathrm{UV}}$ (3.12) and $c_1(R)$ (3.11) independently.

In section 4, we verify all our sum rules in the cases of a free massive boson and a free massive fermion. We find that $c_2(R)$ is monotonic and interpolates between the two central charges in both cases. The divergences associated with massless scalars in de Sitter spoil the sum rule for $c_1(R)$, making it zero for all radii in the free massive boson theory. The

theory of a massless fermion in dS, instead, is devoid of IR divergences and so $c_1(R)$ still interpolates between $c^{\mathrm{UV}}$ and $c^{\mathrm{IR}}$ in the massive fermion flow. We comment on the fact that the massless Schwinger model has the same $c_1$ and $c_2$ functions as the free massive boson theory.

In section 5, we conclude and discuss some open questions.

## 2 The first c-function and its sum rules

In this section we define $c_1(R)$ and we provide sum rules to compute $c^{\mathrm{UV}} - c_1(R)$, checking that the flat space limit reproduces the known formulas from [2, 3]. The techniques we use here closely parallel what was done in [56] to derive sum rules for $c^{\mathrm{UV}}$ in Anti de Sitter.

### 2.1 Preliminaries

We are going to treat both the Euclidean and Lorentzian cases together. The $S^{d+1}$ and $\mathrm{dS}_{d+1}$ can be embedded in $\mathbb{R}^{1,d+1}$ as follows

$$\pm(Y^0)^2 + (Y^1)^2 + \ldots + (Y^{d+1})^2 = R^2 \,. \tag{2.1}$$

We introduce the two-point invariant

$$\sigma \equiv \frac{1}{R^2} Y_1 \cdot Y_2 \tag{2.2}$$

where the dot is either $Y_1 \cdot Y_2 = \eta_{AB} Y_1^A Y_2^B$ or $Y_1 \cdot Y_2 = \delta_{AB} Y_1^A Y_2^B$ depending on the signature of choice, and the indices are $A = 0, \ldots, d+1$. For now, we will set $R = 1$ and then restore it when it is convenient. Operators can be lifted to embedding space and are related to their local counterparts in some coordinates $y^\mu$ with $\mu = 0, 1, \ldots d$ as follows

$$T^{A_1 \ldots A_J} = \frac{\partial Y^{A_1}}{\partial y^{\mu_1}} \cdots \frac{\partial Y^{A_J}}{\partial y^{\mu_J}} T^{\mu_1 \ldots \mu_J} \,. \tag{2.3}$$

The induced metric in embedding space and the covariant derivative are

$$G^{AB} = \eta^{AB} - Y^A Y^B \,, \qquad \nabla^A = \partial_Y^A - Y^A (Y \cdot \partial_Y) \,. \tag{2.4}$$

The proof of the existence of $c_1(R)$ starts from considering the two-point function of the stress tensor on the Bunch-Davies vacuum, with the following choice of normalization[1]

$$T_{\mu\nu} \equiv -\frac{2}{\sqrt{|g|}} \frac{\delta S}{\delta g^{\mu\nu}} \,. \tag{2.5}$$

Let us for now stay in general dimension $d + 1$. By group theory, the two-point function of any spin 2 symmetric tensor can be decomposed into 5 linearly independent tensor structures

$$\langle T^{AB}(Y_1) T^{CD}(Y_2) \rangle = \sum_{i=1}^{5} \mathbb{T}_i^{ABCD} T_i(\sigma) \,. \tag{2.6}$$

---

[1]In our conventions the metric $g_{\mu\nu}$ is dimensionless, so $T_{\mu\nu}$ has mass dimensions $d+1$, as does the trace $\Theta$. Furthermore, we will consider the tensor structures (2.7) dimensionless and the functions $T_i$ dimensionful.

The tensor structures we choose are, specifically,

$$
\begin{aligned}
\mathbb{T}_1^{ABCD} &= V_1^A V_1^B V_2^B V_2^D \,, \\
\mathbb{T}_2^{ACBD} &= V_1^A V_1^C G_2^{BD} + G_1^{AC} V_2^B V_2^D \,, \\
\mathbb{T}_3^{ACBD} &= -V_1^A V_2^B G_{12}^{CD} - V_1^C V_2^D G_{12}^{AB} - V_1^C V_2^B G_{12}^{AD} - V_1^A V_2^D G_{12}^{CB} \,, \\
\mathbb{T}_4^{ACBD} &= G_1^{AC} G_2^{BD} \,, \\
\mathbb{T}_5^{ACBD} &= G_{12}^{CD} G_{12}^{AB} + G_{12}^{AD} G_{12}^{CB} \,.
\end{aligned}
\tag{2.7}
$$

with

$$
\begin{aligned}
V_1^A &= Y_2^A - (Y_1 \cdot Y_2) Y_1^A \,, & V_2^A &= Y_1^A - (Y_1 \cdot Y_2) Y_2^A \,, \\
G_1^{AB} &= \eta^{AB} - Y_1^A Y_1^B \,, & G_2^{AB} &= \eta^{AB} - Y_2^A Y_2^B \,, \\
G_{12}^{AB} &= \eta^{AB} - \frac{Y_2^A Y_1^B}{Y_1 \cdot Y_2} \,.
\end{aligned}
\tag{2.8}
$$

All of the tensors in (2.8) are transverse, so that we force the stress tensor to be tangential to the surface (2.1):

$$
V_i \cdot Y_i = G_i^{AB} Y_{i,A} = G_{12}^{AB} Y_{1,A} = G_{12}^{AB} Y_{2,B} = 0 \,.
\tag{2.9}
$$

The connected two-point function of the trace $\Theta \equiv T_A^A$ will then be given by

$$
\begin{aligned}
\langle \Theta(Y_1) \Theta(Y_2) \rangle =& (\sigma^2 - 1)^2 T_1(\sigma) + 2(d+1)(1 - \sigma^2) T_2(\sigma) + 4 \left( \frac{1}{\sigma} - \sigma \right) T_3(\sigma) \\
& + (d+1) T_4(\sigma) + 2 \left( d + \frac{1}{\sigma^2} \right) T_5(\sigma) \,.
\end{aligned}
\tag{2.10}
$$

The coincident point limit probes the CFT in the UV fixed point of the RG flow defined by our QFT. In particular, in that limit the two-point function (2.6) has to reduce to the CFT two-point function of the stress tensor in the UV. This is uniquely fixed by symmetry and conservation up to a constant that is proportional to $c^{\text{UV}}$. In some Riemann normal coordinates $x^\mu$, this means

$$
\lim_{x \to 0} \langle T^{\mu\nu}(x) T^{\varrho\sigma}(0) \rangle \approx \frac{c_T^{\text{UV}}}{x^{2d+2}} \left[ \frac{1}{2} \left( w_{\mu\varrho} w_{\nu\sigma} + w_{\mu\sigma} w_{\nu\varrho} \right) - \frac{1}{d+1} \eta_{\mu\nu} \eta_{\varrho\sigma} \right] \,,
\tag{2.11}
$$

with

$$
w_{\mu\nu} \equiv \eta_{\mu\nu} - 2 \frac{x_\mu x_\nu}{x^2} \,.
\tag{2.12}
$$

where $c_T^{UV}$ is the normalization of the stress tensor two-point function in the UV, which in two dimensions is related to the central charge as follows

$$
c_T = \frac{1}{2\pi^2} c \,.
\tag{2.13}
$$

This matching in the UV implies that the $T_i(\sigma)$ functions have the following behaviors at coincident points (see appendix (A.2) for more details on how to derive this)

$$
\begin{aligned}
& T_1 \approx \frac{4 c_T^{\text{UV}}}{x^{2d+6}} \,, && T_2 \sim o(x^{-2d-2}) \,, && T_3 \approx -\frac{c_T^{\text{UV}}}{x^{2d+4}} \,, \\
& T_4 \approx -\frac{c_T^{\text{UV}}}{d+1} \frac{1}{x^{2d+2}} \,, && T_5 \approx \frac{c_T^{\text{UV}}}{2} \frac{1}{x^{2d+2}} \,.
\end{aligned}
\tag{2.14}
$$

When defining the stress tensor through (2.5), we effectively impose it to be conserved at the fixed points, but we allow for the presence of local contact terms in its expectation values. To be more precise, (2.14) should also include contact terms in the form of delta functions and their derivatives, such as is done explicitly in [57, 58]. All these terms would drop out of the sum rules we derive, and thus we do not report their explicit forms here.

**Two dimensions** Effectively, in two dimensions (2.6) is a redundant decomposition, since there are only 4 linearly independent tensor structures. This can be seen from the fact that necessarily

$$W^{ABCD}_{EF} \equiv Y_1^{[A} Y_2^B \delta_E^C \delta_F^{D]} = 0 \,, \qquad \text{when } d+1 = 2 \,. \tag{2.15}$$

It is possible to check, then, that the equation $W^{A_1 A_2 CD}_{EF} W^{B_1 B_2 EF}_{CD} = 0$ with $(A_1 B_1)$ and $(A_2 B_2)$ symmetrized, is equivalent to

$$-\frac{2}{\sigma^4} \mathbb{T}_1 - \frac{2}{\sigma^2} \mathbb{T}_2 + \frac{1}{\sigma^3} \mathbb{T}_3 + 2 \frac{1-\sigma^2}{\sigma^2} \mathbb{T}_4 - \frac{1-\sigma^2}{\sigma^2} \mathbb{T}_5 = 0 \,, \tag{2.16}$$

where we suppressed all the indices on the $\mathbb{T}_i$ to avoid clutter. This shows indeed that the tensor structures are degenerate in two dimensions. The $T_i(\sigma)$ functions are then defined up to a common shift by a generic function $g(\sigma)$

$$T_1(\sigma) \sim T_1(\sigma) - \frac{2}{\sigma^4} g(\sigma) \,, \qquad T_2(\sigma) \sim T_2(\sigma) - \frac{2}{\sigma^2} g(\sigma)$$

$$T_3(\sigma) \sim T_3(\sigma) + \frac{1}{\sigma^3} g(\sigma) \,, \qquad T_4(\sigma) \sim T_4(\sigma) - 2 \frac{\sigma^2 - 1}{\sigma^2} g(\sigma) \,, \tag{2.17}$$

$$T_5(\sigma) \sim T_5(\sigma) + \frac{\sigma^2 - 1}{\sigma^2} g(\sigma) \,.$$

We thus construct four quantities which are invariant under this shift[2]

$$\mathcal{T}_1(\sigma) \equiv (1-\sigma^2) \left[ \frac{\sigma^2}{2} T_1(\sigma) - \frac{1}{2} T_2(\sigma) \right] \,, \qquad \mathcal{T}_3(\sigma) \equiv \frac{1}{2} T_4(\sigma) - (1-\sigma^2)\sigma T_3(\sigma)$$

$$\mathcal{T}_2(\sigma) \equiv (1-\sigma^2) \left[ -\frac{1}{2} T_2(\sigma) - \sigma T_3(\sigma) \right] \,, \qquad \mathcal{T}_4(\sigma) \equiv \frac{1}{2} T_4(\sigma) + T_5(\sigma) \,. \tag{2.18}$$

In this basis, the two-point function of the trace of the stress tensor has the following expression, in two dimensions

$$G_\Theta(\sigma) = \frac{2}{\sigma^2} \left( (1-\sigma^2)\mathcal{T}_1(\sigma) - (1+3\sigma^2)\mathcal{T}_2(\sigma) + (3\sigma^2 - 1)\mathcal{T}_3(\sigma) + (1+\sigma^2)\mathcal{T}_4(\sigma) \right) \tag{2.19}$$

where we introduced the notation

$$G_\Theta(\sigma) \equiv \langle \Theta(Y_1)\Theta(Y_2) \rangle \,. \tag{2.20}$$

Notice that the regularity of $G_\Theta(\sigma)$ at $\sigma = 0$, which is not a special configuration on the sphere or in de Sitter, implies the following relation for the $\mathcal{T}_i$ functions

$$\mathcal{T}_1(0) + \mathcal{T}_4(0) = \mathcal{T}_2(0) + \mathcal{T}_3(0) \,. \tag{2.21}$$

---

[2]The precise overall $\sigma$-dependent coefficient of each $\mathcal{T}_i$ function was chosen a posteriori after having derived their spectral decompositions (C.16) in such a way that they would not diverge at antipodal separation $\sigma = -1$.

## 2.2 Proof of the position space sum rule

For the purposes of our proof, we want to find a kernel $r(\sigma)$ and a function $C(\sigma)$ such that

$$r(\sigma)G_\Theta(\sigma) = \frac{d}{d\sigma}C(\sigma)\,. \tag{2.22}$$

Moreover, to extract the central charge, we would like to have $C(1) = c^{\text{UV}}$ up to contact terms, such that integrating both sides of (2.22) will give us a sum rule. For this to work, necessarily $r(1) = 0$ to kill the divergence of $G_\Theta(\sigma)$ at coincident points.

In order to solve (2.22), we use the following ansatz with four unknown functions $g_i(\sigma)$, purely motivated by the fact that it works[3]

$$C(\sigma) = \sum_{i=1}^{4} g_i(\sigma)\mathcal{T}_i(\sigma)\,. \tag{2.23}$$

Then, we impose the conservation of the stress tensor

$$\nabla_A \langle T^{AB}(Y_1)T^{CD}(Y_2)\rangle = 0\,. \tag{2.24}$$

This induces three linearly independent scalar differential equations on the $T_i$ functions, which we obtain by multiplying with three linearly independent projectors, see (A.1) for details. Using (2.18), the conservation equations transform into three differential equations for the $\mathcal{T}_i$'s (A.3).

Call $E_i$ with $i = 1, 2, 3$ the three conservation equations (A.3). Then, we introduce three unknown functions $q_i(\sigma)$ and say

$$r(\sigma)G_\Theta(\sigma) - \frac{d}{d\sigma}C(\sigma) = \sum_{i=1}^{3} q_i(\sigma)E_i\,. \tag{2.25}$$

We impose that this equation be true for any $\mathcal{T}_i(\sigma)$ and any $\mathcal{T}_i'(\sigma)$, giving us 8 differential equations with 8 unknown functions, namely $r(\sigma)$, $g_i(\sigma)$ and $q_i(\sigma)$. We find three solutions which we report in (A.4). Only one of them has $C(1) = c^{\text{UV}}$ up to contact terms and $r(1) = 0$. It has kernel

$$r(\sigma) = 8\pi^2 \left[1 - \sigma\left(\log\left(\frac{1+\sigma}{2}\right) + 1\right)\right]\,, \tag{2.26}$$

and associated function $C(\sigma)$

$$
\begin{aligned}
C(\sigma) = \frac{8\pi^2}{\sigma^2}\Big[ & 2(1-\sigma^2)^2 \log(\zeta)\mathcal{T}_1(\sigma) \\
& + 2(\sigma^2(1-\sigma)^2 + (\sigma^2-1)(2\sigma^2+1)\log(\zeta))\mathcal{T}_2(\sigma) \\
& + (\sigma(1-\sigma)^2(1-2\sigma) + 2(2\sigma^2-1)(1-\sigma^2)\log(\zeta))\mathcal{T}_3(\sigma) \\
& - (\sigma(\sigma-1)^2 + 2(\sigma^2-1)\log(\zeta))\mathcal{T}_4(\sigma)\Big]\,,
\end{aligned}
\tag{2.27}
$$

---

[3]This choice was inspired by [56] where a similar construction led to convergent sum rules for $c^{\text{UV}}$ in AdS$_2$.

where $\zeta \equiv \frac{1+\sigma}{2}$. The fact that $\lim_{\sigma \to 1} C(\sigma) = c^{\mathrm{UV}}$ up to contact terms can be checked by using (2.18) and (2.14). Importantly, at antipodal separation, we have

$$C(-1) = 32\pi^2 R^4 \left( T_5(-1) - T_4(-1) \right) = 32\pi^2 R^4 \left( \mathcal{T}_4(-1) - 3\mathcal{T}_3(-1) \right), \tag{2.28}$$

where we used the fact that the $T_i(\sigma)$ and $\mathcal{T}_i(\sigma)$ functions cannot diverge at $\sigma = -1$ and that $\mathcal{T}_2(-1) = 0$, both facts which we prove in total generality in appendix C.2, and we restored factors of the radius. This is our $c_1(R)$, and we claim it interpolates between $c^{\mathrm{UV}}$ and $c^{\mathrm{IR}}$ as we change the radius of $S^2/\mathrm{dS}_2$

$$c_1(R) \equiv C(-1). \tag{2.29}$$

We will prove that the end-points of $c_1(R)$ are $c^{\mathrm{UV}}$ and $c^{\mathrm{IR}}$, and we will verify in examples in section 4 that $c_1(R)$ is a non-increasing function of the radius in between.

Let us emphasize that, in a given QFT, each $\mathcal{T}_i$ function depends on the mass scales of the theory $\{m_k\}$ through dimensionless products such as $m_k R$ and $m_i/m_j$, hence the dependency on the radius of $c_1(R)$.

Integrating both sides of (2.22) over the domain of the normalized inner product on the sphere $\sigma \in [-1, 1)$, while being careful to avoid contact terms at $\sigma = 1$, we get to one of our main results

$$c^{\mathrm{UV}} - c_1(R) = 8\pi^2 \int_{-1}^{1} d\sigma \left[ 1 - \sigma \left( \log\left( \frac{1+\sigma}{2} \right) + 1 \right) \right] R^4 G_\Theta(\sigma), \tag{2.30}$$

where we restored the necessary factors of the radius.

Let us note that, in this form, this sum rule is analogous to what was obtained in two-dimensional EAdS in [56][4]

$$c^{\mathrm{UV}} = 8\pi^2 \int_{-\infty}^{-1} d\sigma \left[ -1 - \sigma \left( \log\left( \frac{1-\sigma}{2} \right) + 1 \right) \right] R^4 G_\Theta(\sigma), \qquad \text{in AdS}_2. \tag{2.31}$$

Notice the slightly different kernel, the different integration domain and the fact that the information about the intermediate flow is lost in the AdS case.

**Flat space limit** Let us show that (2.30) reduces to (1.1) in the flat space limit, thus proving that $c_1(R)$ interpolates between the two central charges at the fixed points. We start from the flat slicing coordinates $ds^2 = R^2 \frac{-d\eta^2 + dy^2}{\eta^2}$ and we chose conventions in which the metric is dimensionless. The flat space limit is achieved by taking $\eta \to t - R$ and $y \to x$ and then taking $R \to \infty$. Then the metric becomes $ds^2 = -dt^2 + dx^2$ and

$$\sigma = \frac{\eta_1^2 + \eta_2^2 - (y_1 - y_2)^2}{2\eta_1 \eta_2} \sim 1 - \frac{-(t_1 - t_2)^2 + (x_1 - x_2)^2}{2R^2} \equiv 1 - \frac{r^2}{2R^2}. \tag{2.32}$$

---

[4]For a direct comparison, use $\sigma_{\mathrm{here}} = -2\xi_{\mathrm{there}} - 1$.

Our formula (2.30) then changes as follows

$$
\begin{aligned}
c^{\mathrm{UV}} - \lim_{R \to \infty} c_1(R) &= 8\pi^2 \lim_{R \to \infty} \int_{-1}^{1} d\sigma \left[ 1 - \sigma \left( \log \left( \frac{1+\sigma}{2} \right) + 1 \right) \right] R^4 G_\Theta(\sigma) \\
&= 6\pi^2 \lim_{R \to \infty} \int_0^{2R} r^3 dr \, G_\Theta \left( 1 - \frac{r^2}{2R^2} \right) \\
&= 6\pi^2 \int_0^{\infty} r^3 dr \langle \Theta(r)\Theta(0) \rangle^{\mathrm{flat}},
\end{aligned}
\tag{2.33}
$$

The last form precisely matches (1.1), implying that

$$
\lim_{R \to \infty} c_1(R) = c^{\mathrm{IR}}. \tag{2.34}
$$

## 2.3 A sum rule in terms of spectral densities

We are interested in phrasing (2.30) in terms of an integral over the spectrum of the theory. To do that, we are going to use the fact that the two-point function of the trace of the stress tensor in the Bunch-Davies vacuum in a unitary QFT in $\mathrm{dS}_2$ has a Källén-Lehmann decomposition into UIRs of $SO(1,2)$ as follows[5][44–46, 55, 59–62]

$$
G_\Theta(\sigma) = 2\pi \int_{\frac{1}{2}+i\mathbb{R}} \frac{d\Delta}{2\pi i} \, \varrho_\Theta^{\mathcal{P}}(\Delta) G_\Delta(\sigma) + \int_0^1 d\Delta \, \varrho_\Theta^{\mathcal{C}}(\Delta) G_\Delta(\sigma), \tag{2.35}
$$

with

$$
G_\Delta(\sigma) = \frac{1}{4} \csc(\pi\Delta) \, {}_2F_1 \left( \Delta, \bar{\Delta}, 1, \frac{1+\sigma}{2} \right), \qquad \bar{\Delta} \equiv 1 - \Delta, \tag{2.36}
$$

where $\Delta$ parametrizes the eigenvalue of the quadratic Casimir of $SO(1,2)$ as follows

$$
\mathcal{C}_2^{SO(1,2)} = \Delta(1-\Delta) \equiv m^2 R^2. \tag{2.37}
$$

The first term in (2.35) stands for contributions associated to principal series UIRs, while the second one stands for complementary series contributions. In a unitary theory, the spectral densities $\varrho_\Theta^{\mathcal{P}}(\Delta)$ and $\varrho_\Theta^{\mathcal{C}}(\Delta)$ are positive on their domains of integration. For early references on the full classification of UIRs of $SO(1,2)$ see [63–66]. For recent reviews, see [67, 68].

Notice that we are excluding the possibility that, in two dimensions, the discrete series of UIRs ($\Delta = p \in \mathbb{N}/\{0\}$) could contribute to the Källén-Lehmann decomposition of the trace of the stress tensor, since it is a scalar operator. In [45, 50] we phrased more precisely some arguments in favor of the fact that only operators with spin $J \geq p$ can interpolate between the vacuum and states in the discrete series. The results in the examples in section 4 add evidence to this fact, even if a rigorous and complete proof is still missing. Some references which speculate on possible loopholes to these arguments are [69–71]. At the moment, no explicit counterexample to this statement is present in the literature.

---

[5]In our conventions $G_\Delta$ is dimensionless, so the mass dimensions of the trace of the stress tensor are captured by the spectral densities, which thus have mass dimension 4.

Given these assumptions, the derivation of the spectral sum rule is straightforward: we plug (2.35) into (2.30) and carry out the integral over $\sigma$. We use the following identities

$$\int_{-1}^{1} d\sigma \, G_\Delta(\sigma) = \frac{1}{2\pi} \frac{1}{\Delta\bar{\Delta}} \, ,$$

$$\int_{-1}^{1} d\sigma \, \sigma \, G_\Delta(\sigma) = \frac{1}{2\pi} \frac{1}{(\Delta+1)(\bar{\Delta}+1)} \, , \tag{2.38}$$

$$\int_{-1}^{1} d\sigma \, \sigma \log\left(\frac{1+\sigma}{2}\right) G_\Delta(\sigma) = \frac{1}{2\pi} \frac{\Delta\bar{\Delta} - 4}{\Delta\bar{\Delta}(\Delta+1)^2(\bar{\Delta}+1)^2} - \frac{\csc(\pi\Delta)}{2(\Delta+1)(\bar{\Delta}+1)} \, .$$

We obtain

$$c^{\text{UV}} - c_1(R) = \int_{\frac{1}{2}+i\mathbb{R}} \frac{d\Delta}{2\pi i} \left( \frac{24\pi^2}{(\Delta+1)^2(\bar{\Delta}+1)^2} + \frac{8\pi^3 \csc(\pi\Delta)}{(\Delta+1)(\bar{\Delta}+1)} \right) R^4 \varrho_\Theta^\mathcal{P}(\Delta)$$

$$+ \int_0^1 \frac{d\Delta}{2\pi} \left( \frac{24\pi^2}{(\Delta+1)^2(\bar{\Delta}+1)^2} + \frac{8\pi^3 \csc(\pi\Delta)}{(\Delta+1)(\bar{\Delta}+1)} \right) R^4 \varrho_\Theta^\mathcal{C}(\Delta) \, . \tag{2.39}$$

Notice that, now, the integrands on the right hand side are manifestly positive on the principal and complementary series domains $\Delta = \frac{1}{2} + i\mathbb{R}$ and $\Delta \in (0,1)$, implying that

$$c^{\text{UV}} \geq c_1(R) \, . \tag{2.40}$$

**Flat space limit**  Let us show that (2.39) reduces to (1.1) in the flat space limit. We start from the fact that, reinstating factors of the radius, [45]

$$\lim_{R\to\infty} \frac{R}{\sqrt{s}} \varrho_\mathcal{O}^\mathcal{P}(\Delta = iR\sqrt{s}) = \varrho_\mathcal{O}^{\text{flat}}(s) \, , \tag{2.41}$$

where $s \equiv m^2$ is the flat space mass that is integrated over in the Källén-Lehmann representation. Taking the flat space limit of (2.39) gives

$$c^{\text{UV}} - \lim_{R\to\infty} c_1(R) = 12\pi \lim_{R\to\infty} \left[ \int_0^\infty \frac{R ds}{\sqrt{s}} \frac{1}{s^2 R^4} \frac{\sqrt{s}}{R} R^4 \varrho_\Theta^{\text{flat}}(s) + \text{complementary} \right]$$

$$= 12\pi \int_0^\infty \frac{ds}{s^2} \varrho_\Theta^{\text{flat}}(s) + \lim_{R\to\infty} \text{complementary} \tag{2.42}$$

$$= c^{\text{UV}} - c^{\text{IR}} + \lim_{R\to\infty} \text{complementary} \, ,$$

where we used the fact that the $\csc(\pi\Delta)$ factor exponentially suppresses the spectral density in this limit, which cannot compete due to Tauberian theorems in flat space, and that the first term in the penultimate line was exactly the spectral sum rule (1.1). Since we showed that $\lim_{R\to\infty} c_1(R) = c^{\text{IR}}$ in the previous section, we just proved that

$$\lim_{R\to\infty} \int_0^1 \frac{d\Delta}{2\pi} \left( \frac{24\pi^2}{(\Delta+1)^2(\bar{\Delta}+1)^2} + \frac{8\pi^3 \csc(\pi\Delta)}{(\Delta+1)(\bar{\Delta}+1)} \right) R^4 \varrho_\Theta^\mathcal{C}(\Delta) = 0 \, , \tag{2.43}$$

meaning that the complementary series contribution has to vanish when taking the $R \to \infty$ limit.

## 2.4 Behavior of the first c-function at vanishing radius

Let us study the behavior of $c_1(R)$ as we take $R \to 0$. We start from (2.30) and study the limit

$$\lim_{R \to 0} \int_{-1}^{1} d\sigma \left[ 1 - \sigma \left( \log \left( \frac{1+\sigma}{2} \right) + 1 \right) \right] \tilde{G}_\Theta(\sigma, \{m_i R\}),  \tag{2.44}$$

where $\tilde{G}_\Theta \equiv R^4 G_\Theta$ is the dimensionless two-point function of the trace of the stress tensor, and we made explicit the fact that it can in general depend on all the dimensionless combinations of the mass scales of the theory and of the radius. Taking $R \to 0$ in this formula while keeping $m_k$ fixed is equivalent to taking all $m_k \to 0$ and keeping $R$ fixed, probing the UV fixed point of the theory, which is a CFT on a two-sphere of radius $R$. It is a general fact that in a CFT in curved space, the two-point function of the trace of the stress tensor vanishes up to contact terms[6]. Then, we notice two more facts: the divergence of the kernel at $\sigma = -1$ is logarithmic and thus integrable, and the divergence of $\tilde{G}_\Theta$ at $\sigma = 1$ is logarithmic and cured by a simple zero in the kernel. We can thus safely state that

$$\lim_{R \to 0} c_1(R) = c^{\text{UV}}.  \tag{2.45}$$

This has implications regarding the integrals appearing in the spectral sum rule (2.39). Specifically, we can say that necessarily

$$\lim_{R \to 0} \Bigg[ \int_{\frac{1}{2}+i\mathbb{R}} \frac{d\Delta}{2\pi i} \left( \frac{24\pi^2}{(\Delta+1)^2(\bar{\Delta}+1)^2} + \frac{8\pi^3 \csc(\pi\Delta)}{(\Delta+1)(\bar{\Delta}+1)} \right) R^4 \varrho_\Theta^{\mathcal{P}}(\Delta) \\ + \int_0^1 \frac{d\Delta}{2\pi} \left( \frac{24\pi^2}{(\Delta+1)^2(\bar{\Delta}+1)^2} + \frac{8\pi^3 \csc(\pi\Delta)}{(\Delta+1)(\bar{\Delta}+1)} \right) R^4 \varrho_\Theta^{\mathcal{C}}(\Delta) \Bigg] = 0.  \tag{2.46}$$

This will be important when studying the second c-function in the next section.

## 3 Spectral densities of the stress tensor and the second c-function

In this section, we derive a series of properties regarding the spectral representation of the stress tensor in unitary QFTs in de Sitter and on the sphere. In subsection 3.1, we show the most general form of the spectral decomposition of the stress tensor in $d+1$ dimensions, taking into consideration only the contributions from the principal and complementary series. The conservation of the stress tensor implies relations between the spectral densities associated to different $SO(d)$ spin, making the expressions much simpler than initially expected. In subsection 3.2, we specify to the case of two dimensions and we take into account all the UIRs that the stress tensor can in principle couple to. After imposing conservation, we find once again a much more compact expression than one would expect, and we show that from the discrete series only the $\Delta = 2$ UIR can appear. We leave many details of this section to the appendix C.1.

---

[6]The precise expression is $\langle \Theta(x_1)\Theta(x_2) \rangle = -\frac{c}{12\pi} \nabla^2 \delta^{(2)}(x_1 - x_2)$ [57]

### 3.1 Spectral decomposition of the stress tensor in higher dimensions

The stress tensor is a symmetric spin 2 operator. As such, *naively*, one would expect it to have a total of five independent spectral densities: one associated to its trace, three associated to the $SO(d)$ decomposition of its traceless part, and one associated to the mixed two-point function of its trace and its traceless parts. In equations,

$$\langle T^{AB}(Y_1)T^{CD}(Y_2)\rangle = 2\pi \int_{\frac{d}{2}+i\mathbb{R}} \frac{d\Delta}{2\pi i}\Bigg[ \sum_{\ell=0}^{2} \varrho_{\hat{T},\ell}^{\mathcal{P}}(\Delta)G_{\Delta,\ell}^{AB,CD}(Y_1,Y_2)$$
$$+ \varrho_{\hat{T}\Theta}^{\mathcal{P}}(\Delta)\left( \frac{G_2^{CD}}{d+1}\hat{\Pi}_1^{AB}G_\Delta(\sigma) + \frac{G_1^{AB}}{d+1}\hat{\Pi}_2^{CD}G_\Delta(\sigma)\right)$$
$$+ \varrho_\Theta^{\mathcal{P}}(\Delta)\frac{G_1^{AB}G_2^{CD}}{(d+1)^2}G_\Delta(\sigma)\Bigg] + \text{other UIRs}. \qquad (3.1)$$

where $G_{\Delta,\ell}^{AB,CD}(Y_1,Y_2)$ are the blocks that appear in the Källén-Lehmann representation of traceless symmetric spin 2 operators on the Bunch-Davies vacuum, $\hat{\Pi}_i^{AB}$ is a traceless symmetric differential operator

$$\hat{\Pi}_i^{AB} = \frac{1}{d+1}G_i^{AB}\nabla_i^2 - \nabla_i^{(A}\nabla_i^{B)}, \qquad (3.2)$$

and $G_\Delta(\sigma)$ is the canonically normalized free scalar propagator in $d+1$ dimensions in the Bunch-Davies vacuum[7]

$$G_\Delta(\sigma) = \frac{\Gamma(\Delta)\Gamma(\bar{\Delta})}{(4\pi)^{\frac{d+1}{2}}}\mathbf{F}\left(\Delta,\bar{\Delta},\frac{d+1}{2},\frac{1+\sigma}{2}\right), \qquad \sigma = \frac{1}{R^2}Y_1\cdot Y_2. \qquad (3.3)$$

In particular, the $\ell = 2$ block $G_{\Delta,2}^{AB,CD}(Y_1,Y_2)$ is explicitly reported in (B.16) and the ones for $\ell = 0$ and $\ell = 1$ can be found in index-free form in appendix F.3 of [45]. Embedding space covariant derivatives $\nabla^A$ and the induced metric $G^{AB}$ are defined in section 2.1. In this section, we are again setting $R = 1$.

Taking or removing traces from (3.1) reduces it to the following, naively independent, decompositions

$$\langle \hat{T}^{AB}(Y_1)\hat{T}^{CD}(Y_2)\rangle = 2\pi \int_{\frac{d}{2}+i\mathbb{R}} \frac{d\Delta}{2\pi i} \sum_{\ell=0}^{2} \varrho_{\hat{T},\ell}^{\mathcal{P}}(\Delta)G_{\Delta,\ell}^{AB,CD}(Y_1,Y_2) + \text{other UIRs},$$

$$\langle \Theta(Y_1)\hat{T}^{CD}(Y_2)\rangle = 2\pi \int_{\frac{d}{2}+i\mathbb{R}} \frac{d\Delta}{2\pi i}\ \varrho_{\hat{T}\Theta}^{\mathcal{P}}(\Delta)\hat{\Pi}_2^{CD}G_\Delta(\sigma) + \text{other UIRs}, \qquad (3.4)$$

$$\langle \Theta(Y_1)\Theta(Y_2)\rangle = 2\pi \int_{\frac{d}{2}+i\mathbb{R}} \frac{d\Delta}{2\pi i}\ \varrho_\Theta^{\mathcal{P}}(\Delta)G_\Delta(\sigma) + \text{other UIRs}.$$

The conservation of the stress tensor induces relations among these spectral densities, totally analogous to those in flat space [5]. We relegate the proof of these relations to

---

[7]We use the notation for the regularized hypergeometric function $\mathbf{F}(a,b,c,z) \equiv \frac{1}{\Gamma(c)}{}_2F_1(a,b,c,z)$, and in this section $\bar{\Delta} \equiv d - \Delta$, while in the rest of the paper $\bar{\Delta} \equiv 1 - \Delta$.

appendix C.1. Here, we report the results[8]

$$\varrho_{\hat{T}\Theta}(\Delta) = \frac{\varrho_\Theta(\Delta)}{d(\Delta+1)(\bar{\Delta}+1)}\,, \quad \varrho_{\hat{T},0}(\Delta) = \frac{\varrho_\Theta(\Delta)}{d^2(\Delta+1)^2(\bar{\Delta}+1)^2}\,, \quad \varrho_{\hat{T},1}(\Delta) = 0\,. \quad (3.5)$$

The Källén-Lehmann decomposition of the stress tensor thus reduces to

$$\langle T^{AB}(Y_1)T^{CD}(Y_2)\rangle = 2\pi \int_{\frac{d}{2}+i\mathbb{R}} \frac{d\Delta}{2\pi i}\Big[\varrho^{\mathcal{P}}_{\hat{T},2}(\Delta)G^{AB,CD}_{\Delta,2}(Y_1,Y_2)$$
$$+ \frac{\varrho^{\mathcal{P}}_\Theta(\Delta)}{d^2(\Delta+1)^2(\bar{\Delta}+1)^2}\Pi_1^{AB}\Pi_2^{CD}G_\Delta(\sigma)\Big] \qquad (3.6)$$
$$+ \text{other UIRs}$$

where $\Pi_i^{AB}$ comes from the combination of the various propagators proportional to $\varrho_\Theta$ after applying (3.5)

$$\Pi_i^{AB} \equiv G_i^{AB}\left(d+\nabla_i^2\right) - \nabla_i^{(A}\nabla_i^{B)}\,. \qquad (3.7)$$

In this representation, both lines in (3.6) are independently conserved: $\nabla_A G^{AB,CD}_{\Delta,2} = 0$ by definition and it can be checked that $\nabla_A \Pi^{AB}G_\Delta = 0$. Group theoretically, the first line corresponds to states which carry $SO(d)$ spin 2, while the second line corresponds to all other scalar states.

## 3.2 Spectral decomposition of the stress tensor in two dimensions

In two dimensions, the picture simplifies even further: there is no dynamical propagating massive traceless symmetric spin 2 field, so $G^{AB,CD}_\Delta(Y_1,Y_2) = 0$. Moreover, the only UIRs that can contribute, other than the principal series, are the complementary series and the irrep with $\Delta = 2$ in the discrete series. We prove this fact in appendix C.1. We are left with[9]

$$\langle T^{AB}(Y_1)T^{CD}(Y_2)\rangle = 2\pi \int_{\frac{1}{2}+i\mathbb{R}} \frac{d\Delta}{2\pi i}\frac{\varrho^{\mathcal{P}}_\Theta(\Delta)}{(\Delta+1)^2(\bar{\Delta}+1)^2}\Pi_1^{AB}\Pi_2^{CD}G_\Delta(\sigma)$$
$$+ \int_0^1 d\Delta \frac{\varrho^{\mathcal{C}}_\Theta(\Delta)}{(\Delta+1)^2(\bar{\Delta}+1)^2}\Pi_1^{AB}\Pi_2^{CD}G_\Delta(\sigma) \qquad (3.8)$$
$$+ \varrho^{\mathcal{D}_2}_{\hat{T}}\Pi_1^{AB}\Pi_2^{CD}G_{\Delta=2}(\sigma)\,.$$

---

[8]Here we omit the superscripts on the spectral densities specifying the series of UIRs because these identities apply also to the complementary series, given that the functional form of its contribution is just the analytic continuation of the principal series ones.

[9]We use the same notation for projectors and propagators that we used in the higher dimensional case, but we are implicitly setting $d = 1$.

Since for the discrete series there is no integral over $\Delta$, we call $\varrho_{\hat{T}}^{\mathcal{D}_2}$ the spectral weight of the stress tensor in the $\Delta = 2$ UIR.

When the theory has a good continuation in the number of spacetime dimensions, there is a final simplification. If in higher dimensions the stress tensor only decomposes in principal and complementary series, then when continuing to $d = 1$ the only contribution to the $\Delta = 2$ discrete series comes from spurious poles at $\Delta = 2$ and $\bar{\Delta} = 2$ in $G_{\Delta,2}^{AB,CD}$ which will cross the contour of integration over the principal series and lead to the discrete series $\Delta = 2$ contribution. This is in fact what happens in the free massive boson case, as we will discuss in further detail in section 4.1 and appendix B.1. We can thus state that if the theory has a good analytic continuation in $d$, with only principal and complementary series contributions to the stress tensor in higher dimensions, we have

$$\varrho_{\hat{T}}^{\mathcal{D}_2} = 4\pi \operatorname*{Res}_{\Delta=2} \left( \frac{\varrho_{\Theta}^{\mathcal{P}}(\Delta)}{(\Delta+1)^2(\bar{\Delta}+1)^2} \right) . \tag{3.9}$$

## 3.3 Finding the second c-function

By comparing the definitions (2.6), (2.18) and (2.29) with the spectral decomposition (3.8), it is possible to derive formulas which extract $c^{\mathrm{UV}}$ and $c_1(R)$ individually as integrals over the spectral densities of the stress tensor. To start, in appendix C.2 we show how to relate the $\mathcal{T}_i(\sigma)$ functions to integrals over the spectral densities of the stress tensor, obtaining equations (C.16). Then, evaluating them at $\sigma = -1$, we obtain

$$\mathcal{T}_1(-1) = 0 , \qquad\qquad\qquad\qquad \mathcal{T}_2(-1) = 0 ,$$
$$\mathcal{T}_3(-1) = -\frac{3}{32\pi}\varrho_{\hat{T}}^{\mathcal{D}_2} + \frac{\pi}{32}\int_{\frac{1}{2}+i\mathbb{R}} \frac{d\Delta}{2\pi i} \frac{(4+\Delta\bar{\Delta})\csc(\pi\Delta)}{(\Delta+1)(\bar{\Delta}+1)} \varrho_{\Theta}^{\mathcal{P}}(\Delta) + \text{complementary} ,$$
$$\mathcal{T}_4(-1) = \frac{3}{32\pi}\varrho_{\hat{T}}^{\mathcal{D}_2} + \frac{\pi}{32}\int_{\frac{1}{2}+i\mathbb{R}} \frac{d\Delta}{2\pi i} \frac{(4+3\Delta\bar{\Delta})\csc(\pi\Delta)}{(\Delta+1)(\bar{\Delta}+1)} \varrho_{\Theta}^{\mathcal{P}}(\Delta) + \text{complementary} , \tag{3.10}$$

where "complementary" stands for the same exact expression as the principal series case but with an integral over the $\Delta \in (0,1)$ contour. Now, using the definition of $c_1(R)$ (2.29), we get

$$c_1(R) = 12\pi R^4 \left( \varrho_{\hat{T}}^{\mathcal{D}_2} - \frac{2\pi^2}{3}\int_{\frac{1}{2}+i\mathbb{R}} \frac{d\Delta}{2\pi i} \frac{\csc(\pi\Delta)\varrho_{\Theta}^{\mathcal{P}}(\Delta)}{(\Delta+1)(\bar{\Delta}+1)} - \frac{\pi}{3}\int_0^1 d\Delta \frac{\csc(\pi\Delta)\varrho_{\Theta}^{\mathcal{C}}(\Delta)}{(\Delta+1)(\bar{\Delta}+1)} \right) . \tag{3.11}$$

Using our sum rule (2.39), we can thus derive a formula for $c^{\mathrm{UV}}$ which is valid for any $R$:

$$c^{\mathrm{UV}} = 12\pi R^4 \left( \varrho_{\hat{T}}^{\mathcal{D}_2} + 2\pi\int_{\frac{1}{2}+i\mathbb{R}} \frac{d\Delta}{2\pi i} \frac{\varrho_{\Theta}^{\mathcal{P}}(\Delta)}{(\Delta+1)^2(\bar{\Delta}+1)^2} + \int_0^1 d\Delta \frac{\varrho_{\Theta}^{\mathcal{C}}(\Delta)}{(\Delta+1)^2(\bar{\Delta}+1)^2} \right) . \tag{3.12}$$

Interestingly, in the flat space limit the second term in (3.12) independently reduces to the sum rule for $c^{\mathrm{UV}} - c^{\mathrm{IR}}$, see the previous paragraph. At the same time, the principal series integral in (3.11) vanishes in this limit. Moreover, (2.43) implies that both complementary series integrals in (3.12) and (3.11) vanish in this limit. Finally, in section 2.4 we showed

that all of these integrals of the spectral densities of the trace of the stress tensor vanish as $R \to 0$. We can thus define

$$c_2(R) \equiv 12\pi R^4 \varrho_{\hat{T}}^{\mathcal{D}_2} \,, \tag{3.13}$$

and state that

$$\begin{aligned}
\lim_{R \to \infty} c_2(R) &= c^{\text{IR}} \,, \\
\lim_{R \to 0} c_2(R) &= c^{\text{UV}} \,.
\end{aligned} \tag{3.14}$$

In other words, the spectral weight of the stress tensor in the discrete series $\Delta = 2$ irrep is another candidate $c$-function which interpolates between $c^{\text{IR}}$ and $c^{\text{UV}}$ as we vary the radius. In the example of the free boson, we explicitly checked that it is also monotonic for intermediate radii. Let us write down two sum rules for $c_2(R)$. The one in terms of spectral densities is obtained by combining (3.11), (3.12) and (3.13):

$$c^{\text{UV}} - c_2(R) = 24\pi^2 R^4 \left( \int_{\frac{1}{2}+i\mathbb{R}} \frac{d\Delta}{2\pi i} \frac{\varrho_{\Theta}^{\mathcal{P}}(\Delta)}{(\Delta+1)^2(\bar{\Delta}+1)^2} + \int_0^1 \frac{d\Delta}{2\pi} \frac{\varrho_{\Theta}^{\mathcal{C}}(\Delta)}{(\Delta+1)^2(\bar{\Delta}+1)^2} \right) \tag{3.15}$$

once again from this we can deduce $c^{\text{UV}} \geq c^{\text{IR}}$. Comparing with (2.39) we can further state

$$c_2(R) \geq c_1(R) \,. \tag{3.16}$$

Deriving the position space sum rule for $c_2(R)$ is slightly more involved. We make use of the inversion formula from [39, 45, 46], which in two dimensions states that the principal series spectral density associated to a two-point function $G(\sigma)$ is given by

$$\rho^{\mathcal{P}}(\Delta) = \left( \frac{1}{2} - \Delta \right) i \cot(\pi\Delta) \int_{\mathcal{C}_k} d\sigma \; {}_2F_1\left( \Delta, \bar{\Delta}, 1, \frac{1-\sigma}{2} \right) G(\sigma) \,, \tag{3.17}$$

with the contour $\mathcal{C}_k$ being a "keyhole" contour wrapping the branch cut of $G(\sigma)$, which for a physical two-point function is at $\sigma \in [1, \infty)$, see figure 3. In practice, evaluating this integral corresponds to computing the residue of the integrand at $\sigma = 1$ and the discontinuity of $G(\sigma)$ around the cut.

For now, we will assume that there are no further contributions to the spectral decomposition of our two-point function. We will see that the sum rule we obtain in this way works even if there are complementary series contributions.

To proceed, we plug (3.17) inside (3.15), and we use the following identity

$${}_2F_1\left( \Delta, \bar{\Delta}, 1, \frac{1-\sigma}{2} \right) = \frac{\Gamma(\frac{1}{2}-\Delta)}{\sqrt{\pi}\Gamma(\bar{\Delta})} \frac{1}{(2(1+\sigma))^{\Delta}} \; {}_2F_1\left( \Delta, \Delta, 2\Delta, \frac{2}{1+\sigma} \right) + (\Delta \leftrightarrow \bar{\Delta}) \,. \tag{3.18}$$

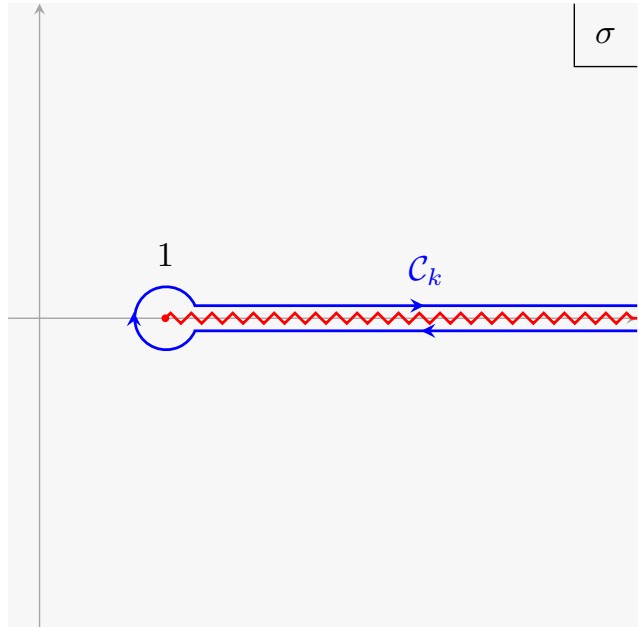

**Figure 3:** In blue, the contour of integration $\mathcal{C}_k$ in (3.17). It wraps the branch cut of the two-point function for time-like separation. In practice, it is equivalent to summing the residue at $\sigma = 1$ and the discontinuity at $\sigma \in (1, \infty)$.

Exploiting the symmetry of the integral, we can drop the second term in (3.18). Now the integrand decays with large $\text{Re}(\Delta)$. Assuming the convergence of the sum rule (3.15) and of the inversion formula (3.17), we can swap the integrals and close the contour of integration over $\Delta$ on the right half of the complex plane, picking up the residue on the only pole of the integrand, which is at $\Delta = 2$. We obtain the following position space sum rule,

$$c^{\text{UV}} - c_2(R) = \int_{\mathcal{C}_k} d\sigma \; r_2(\sigma) R^4 G_\Theta(\sigma) \,. \tag{3.19}$$

with the explicit form of the kernel being

$$r_2(\sigma) = \frac{4\pi i}{(1+\sigma)^2} \left[ 2\sigma^3 \coth^{-1}(\sigma) + \sigma^2 \log\left(\frac{(\sigma+1)^3}{4(\sigma-1)}\right) + \sigma \log\left(\frac{(\sigma-1)(\sigma+1)^3}{16}\right) \right. \tag{3.20}$$

$$\left. + \log\left(\frac{\sigma^2 - 1}{4}\right) + 2(1+\sigma)^2 \left(1 - \sigma\left(\coth^{-1}(\sigma)\log\left(\frac{\sigma-1}{2}\right) - \text{Li}_2\left(\frac{2}{1-\sigma}\right)\right)\right) \right],$$

where $\text{Li}_2(x)$ is a dilogarithm. Notice that $r_2(\sigma)$ is purely imaginary in $\sigma \in (1, \infty)$, as is the discontinuity of $G_\Theta(\sigma)$, so that the integrand in (3.19) is real.

Let us also state that, as long as the two-point function $G_\Theta(\sigma)$ can be analytically continued to some regime where only the principal series contributes to its spectral decomposition, then this formula works in every other regime. We checked this works in the free massive boson case, even when complementary series contributions appear.

As a final note, let us emphasize that what we showed in this section implies that in any QFT[10] the spectral decomposition of the stress tensor *must* contain a contribution from the $\Delta = 2$ discrete series irrep, since its spectral weight has to interpolate between $c^{\text{IR}}$ and $c^{\text{UV}}$.

### 3.4 Independent argument for the second c-function

Here we will give an independent argument for why $\varrho_{\hat{T}}^{\mathcal{D}_2}$ interpolates between $c^{\text{UV}}$ and $c^{\text{IR}}$ as we tune the radius of the sphere $R$. Let us start by writing down the Källén-Lehmann decomposition of the stress tensor in two-dimensional flat space [3–5]

$$\langle T^{\mu\nu}(x_1)T^{\rho\sigma}(x_2)\rangle^{\text{flat}} = \frac{c^{\text{IR}}}{12\pi}\Pi_1^{\mu\nu}\Pi_2^{\rho\sigma}G_0(x_1, x_2) + \int_0^\infty \frac{ds}{s^2}\tilde{\varrho}_\Theta(s)\Pi_1^{\mu\nu}\Pi_2^{\rho\sigma}G_s(x_1, x_2)\,, \quad (3.21)$$

where we separated the massless contributions from the massive ones, and

$$G_s(x_1, x_2) \equiv \frac{1}{2\pi}K_0(\sqrt{s}|x_1 - x_2|)\,, \quad (3.22)$$

is the canonically normalized propagator of a massive free scalar with $m^2 = s$ in two dimensions, with $K_n(x)$ being the modified Bessel function of the second kind, and

$$\Pi_i^{\mu\nu} \equiv \eta^{\mu\nu}\partial_i^2 - \partial_i^\mu\partial_i^\nu\,, \quad (3.23)$$

are the divergence-less projectors which ensure conservation of the stress tensor. Notice that the massless contribution in (3.21) is also traceless. That is necessary, since it is what survives in the IR CFT. In fact, it can be checked that

$$\frac{c^{\text{IR}}}{12\pi}\Pi_1^{\mu\nu}\Pi_2^{\rho\sigma}G_0(x_1, x_2) = \langle T^{\mu\nu}(x_1)T^{\rho\sigma}(x_2)\rangle_{\text{CFT}}^{\text{flat}}\,. \quad (3.24)$$

On the other hand, consider the Källén-Lehmann decomposition of the stress tensor in $S^2/\text{dS}_2$ which we derived in the previous section and which we report here for convenience

$$\begin{aligned}\langle T^{AB}(Y_1)T^{CD}(Y_2)\rangle = &2\pi\int_{\frac{1}{2}+i\mathbb{R}}\frac{d\Delta}{2\pi i}\frac{\varrho_\Theta^{\mathcal{P}}(\Delta)}{(\Delta+1)^2(\bar{\Delta}+1)^2}\Pi_1^{AB}\Pi_2^{CD}G_\Delta(\sigma)\\&+\int_0^1 d\Delta\frac{\varrho_\Theta^{\mathcal{C}}(\Delta)}{(\Delta+1)^2(\bar{\Delta}+1)^2}\Pi_1^{AB}\Pi_2^{CD}G_\Delta(\sigma)\\&+\varrho_{\hat{T}}^{\mathcal{D}_2}\Pi_1^{AB}\Pi_2^{CD}G_{\Delta=2}(\sigma)\,.\end{aligned} \quad (3.25)$$

In [45] we studied the flat space limit of the principal series contributions and showed that they account for the continuum part in (3.21). Then, in (2.43) we argued that the complementary series part has to vanish in the flat space limit. What remains is only the last line in (3.25). Now notice that the $\Delta = 2$ contribution is precisely the two-point function of the stress tensor in a CFT on the two-sphere, up to a normalization factor

$$\begin{aligned}W_{1A}^\pm W_{1B}^\pm W_{2C}^\pm W_{2D}^\pm\Pi_1^{AB}\Pi_2^{CD}G_{\Delta=2}(\sigma) &= \frac{6}{\pi}\frac{(W_1^\pm \cdot W_2^\pm)^2}{(1-\sigma)^4}\\&\propto \langle T(Y_1, W_1^\pm)T(Y_2, W_2^\pm)\rangle_{\text{CFT}}^{\text{sphere}}\,,\end{aligned} \quad (3.26)$$

---

[10]The only exception is, of course, the empty theory.

where $W_{iA}^{\pm}$ are null vectors we are using to contract indices and give a compact form to the final expression, and the $\pm$ stands for their behavior under parity. We explain some more details on them in appendix B.1 and in our previous work [45]. Then, based on what we argued about the flat space limit, this is what matches the massless part in (3.21) when $R \to \infty$, so that

$$c^{\text{IR}} = 12\pi \lim_{R \to \infty} R^4 \varrho_{\hat{T}}^{\mathcal{D}_2} . \tag{3.27}$$

On the other hand, as we discussed in 2.4, taking $R \to 0$ is equivalent to probing the theory on the sphere at fixed radius but with all mass scales taken to zero, effectively flowing to the UV CFT on $S^2/\text{dS}_2$, where the spectral densities of the trace of the stress tensor vanish and the only term surviving in (3.25) is the discrete series. This implies that

$$c^{\text{UV}} = 12\pi \lim_{R \to 0} R^4 \varrho_{\hat{T}}^{\mathcal{D}_2} , \tag{3.28}$$

giving an independent argument for why $c_2(R)$ defined in (3.13) interpolates between $c^{\text{UV}}$ and $c^{\text{IR}}$.

## 4 Examples

In this section, we apply the sum rules (2.30), (2.39), (3.15) and (3.19) in the cases of a free massive scalar and a free massive fermion to compute the associated c-functions $c_1(R)$ and $c_2(R)$. In the free massive boson case we compute all the spectral densities of the stress tensor and show that the conservation relations (3.5) are satisfied.

### 4.1 Free massive scalar

Consider the theory of a free massive scalar with $m^2 R^2 = \Delta_\phi(1 - \Delta_\phi)$.

$$S = -\frac{1}{2} \int d^2x \sqrt{g} \left( g^{\mu\nu} \partial_\mu \phi \partial_\nu \phi + m^2 \phi^2 \right) , \tag{4.1}$$

In the UV, this can be seen as the free theory of a massless scalar, for which we expect $c^{\text{UV}} = 1$, perturbed by the relevant operator $m^2 \phi^2$. Following the flow to the IR, we get to the trivial empty theory, $c^{\text{IR}} = 0$. In flat space, this is one of the simplest examples of RG flows in QFT and the sum rules (1.1) work perfectly fine. In de Sitter, the IR divergences associated to the zero mode of a massless scalar will instead slightly spoil this picture.

As we take the radius to zero, in fact, we are going to find that the two-point function of the trace of the stress tensor becomes a non-zero constant, due to the divergence of $\langle \phi\phi \rangle$ in this limit, which is equivalent to the massless limit. This is going to affect $c_1(R)$, which is inherently connected to the trace of the stress tensor, and will simply be zero for all radii. Instead, $c_2(R)$, which only depends on the traceless part of the stress tensor, will succesfully interpolate between $c^{\text{UV}}$ and $c^{\text{IR}}$. Let us discuss the details.

The stress tensor for this theory, computed from its definition (2.5), is

$$T_{\mu\nu} = \partial_\mu \phi \partial_\nu \phi - \frac{1}{2} g_{\mu\nu} \left[ \partial^\rho \phi \partial_\rho \phi + m^2 \phi^2 \right] . \tag{4.2}$$

Its trace is $\Theta = -m^2\phi^2$. The two-point function of the trace is thus

$$G_\Theta(\sigma) = 2m^4 \left(G_{\Delta_\phi}(\sigma)\right)^2 . \tag{4.3}$$

Using the explicit expression of $G_\Delta(\sigma)$ (2.36), the sum rule reads

$$c^{\mathrm{UV}} - c_1(R) = 16\pi^2 \int_{-1}^{1} d\sigma \left[1 + \sigma\left(\log\left(\frac{2}{1+\sigma}\right) - 1\right)\right] m^4 R^4 \left(G_{\Delta_\phi}(\sigma)\right)^2 . \tag{4.4}$$

This integral can be carried out numerically and, to arbitrary precision, it returns

$$c^{\mathrm{UV}} - c_1(R) = 1 , \tag{4.5}$$

implying that $c_1(R) = 0$ for all $R$. In appendix B.1, we compute the full two-point function of $T^{\mu\nu}$ and independently verify that $c_1(R) = 0$ using its definition (2.29). This is due to the fact that the massless scalar theory is ill defined in de Sitter, because of the divergent zero mode. This leads to the fact that, for example, the two-point function of the trace of the stress tensor does not vanish as $R \to 0$, but rather it asymptotes to a constant.

$$\lim_{R\to 0} 2m^4 R^4 \left(G_{\Delta_\phi}(\sigma)\right)^2 = \frac{1}{8\pi^2} . \tag{4.6}$$

While this problem affects our first c-function $c_1(R)$, which depends on the trace of the stress tensor, it does not affect $c_2(R)$, which only depends on its traceless part.

In order to study $c_2(R)$ we first need to discuss the spectral decomposition of the stress tensor in this theory. In appendix B.1 we compute all the spectral densities and we check formulas (3.5) and (2.39). Let us report here the resulting decomposition

$$\langle T^{AB}(Y_1) T^{CD}(Y_2)\rangle = 2\pi \int_{\frac{1}{2}+i\mathbb{R}} \frac{d\Delta}{2\pi i} \frac{\varrho_\Theta^{\mathcal{P}}(\Delta)}{(\Delta+1)^2(\bar\Delta+1)^2} \Pi_1^{AB}\Pi_2^{CD} G_\Delta(\sigma)$$

$$+ \theta\left(\mathrm{Re}\Delta_\phi - \frac{3}{4}\right) \int_0^1 d\Delta \frac{\varrho_\Theta^{\mathcal{C}}(\Delta)}{(\Delta+1)^2(\bar\Delta+1)^2} \Pi_1^{AB}\Pi_2^{CD} G_\Delta(\sigma) + (\Delta_\phi \to 1 - \Delta_\phi)$$

$$+ \varrho_{\hat T}^{\mathcal{D}_2} \Pi_1^{AB}\Pi_2^{CD} G_{\Delta=2}(\sigma) , \tag{4.7}$$

where $\theta(x)$ is a Heaviside step function, signaling the appearance of a complementary series irrep if the free scalar is light enough, and the differential operators $\Pi_i^{AB}$ where defined in (3.7). As expected from our arguments, we observe the presence of a discrete series irrep with $\Delta = 2$. The explicit form of the spectral densities is, for the principal series

$$\varrho_\Theta^{\mathcal{P}}\left(\frac{1}{2} + i\lambda\right) = \frac{m^4 \lambda \sinh(\pi\lambda)}{16\pi^4 \Gamma(\frac{1}{2} \pm i\lambda)} \Gamma\left(\frac{\frac{1}{2} \pm i\lambda}{2}\right)^2 \prod_{\pm,\pm} \Gamma\left(\frac{\frac{1}{2} \pm i\lambda \pm 2i\lambda_\phi}{2}\right) , \tag{4.8}$$

where we used $\Delta_\phi = \frac{1}{2} + i\lambda_\phi$ for convenience, so then $m^2 R^2 = \frac{1}{4} + \lambda_\phi^2$. For the complementary and discrete series we find

$$\varrho_\Theta^{\mathcal{C}}(\Delta) = -\delta(\Delta - 2\Delta_\phi + 1) \frac{(\Delta+1)^2 \bar\Delta \cos(\pi\Delta) \Gamma(\frac{3}{2} - \Delta) \Gamma(\frac{3-\Delta}{2}) \Gamma(\frac{\Delta}{2})^2}{2^{4-\Delta} \pi^2 R^4 \Gamma(1 - \frac{\Delta}{2})} ,$$

$$\varrho_{\hat T}^{\mathcal{D}_2} = \frac{\lambda_\phi m^2}{3R^2} \mathrm{csch}(2\pi\lambda_\phi) , \tag{4.9}$$

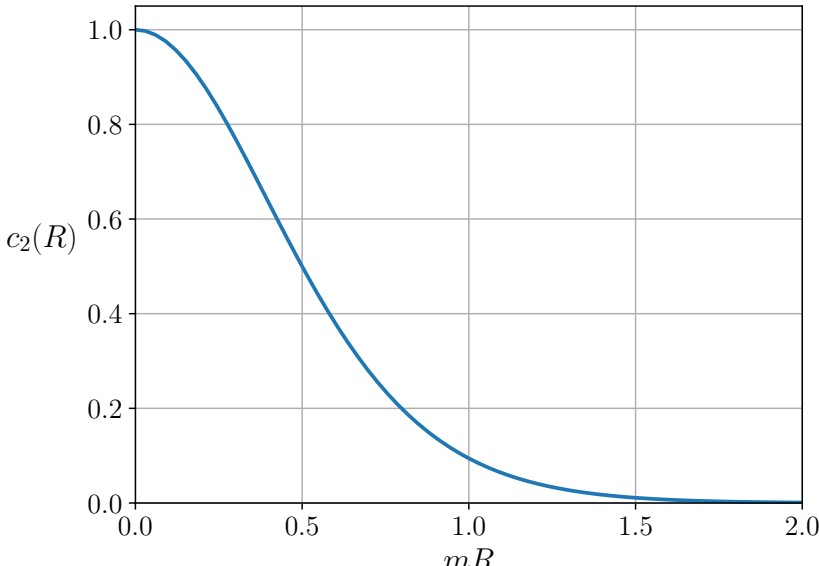

**Figure 4:** Plot of the second c-function in the free massive scalar case, for which we derived an analytic expression, eq. (4.11). It interpolates between $c^{\mathrm{UV}} = 1$, the CFT of the free massless scalar, and $c^{\mathrm{IR}} = 0$, the empty theory.

which can be checked to be positive[11]. If one considers a massless and compact scalar, as in [38], then $G_\Theta(\sigma) = \varrho_\Theta^{\mathcal{P}}(\Delta) = 0$ and $\varrho_{\hat{T}}^{\mathcal{D}_2} = \frac{1}{12\pi R^4}$, giving

$$\langle T^{AB}(Y_1)T^{CD}(Y_2)\rangle = \frac{1}{12\pi R^4}\Pi_1^{AB}\Pi_2^{CD}G_{\Delta=2}(\sigma)\,. \tag{4.10}$$

In the massless case the stress tensor precisely corresponds to the $\Delta = 2$ irrep in the discrete series. This makes sense, given that this theory is conformally invariant and the stress tensor in a CFT is a spin 2 primary with $\Delta = 2$.

Now that we have all the spectral densities, we can check the individual formulas for $c^{\mathrm{UV}}$ (3.12) and $c_1(R)$ (3.11). We find once again that $c^{\mathrm{UV}} = 1$ and that $c_1(R) = 0$ for all $R$, due to the IR issues of the massless scalar theory in de Sitter. We can also compute the second c-function $c_2(R)$ from its definition (3.13), and we obtain explicitly

$$c_2(R) = 4\pi m^2 R^2\sqrt{m^2 R^2 - \frac{1}{4}}\,\mathrm{csch}\left(2\pi\sqrt{m^2 R^2 - \frac{1}{4}}\right)\,. \tag{4.11}$$

As argued before, this function depends only on the traceless part of the stress tensor and is insensitive to the IR divergence of the massless scalar theory. We plot it in figure 4, and we observe that it indeed is a monotonic function which interpolates between $c^{\mathrm{UV}} = 1$ and $c^{\mathrm{IR}} = 0$. We also check that the sum rule (3.19) returns the same function, testing the fact that it works even when the complementary series contributes to the spectral decomposition of $G_\Theta(\sigma)$.

---

[11]The density $\varrho_\Theta^{\mathcal{D}_2}$ is positive for all $\lambda_\phi \in \mathbb{R} \cup i(-\frac{1}{2}, \frac{1}{2})$. The complementary series density $\varrho_\Theta^{\mathcal{C}}$ is positive on the support of the Heaviside theta function in (4.7) after applying the Dirac delta

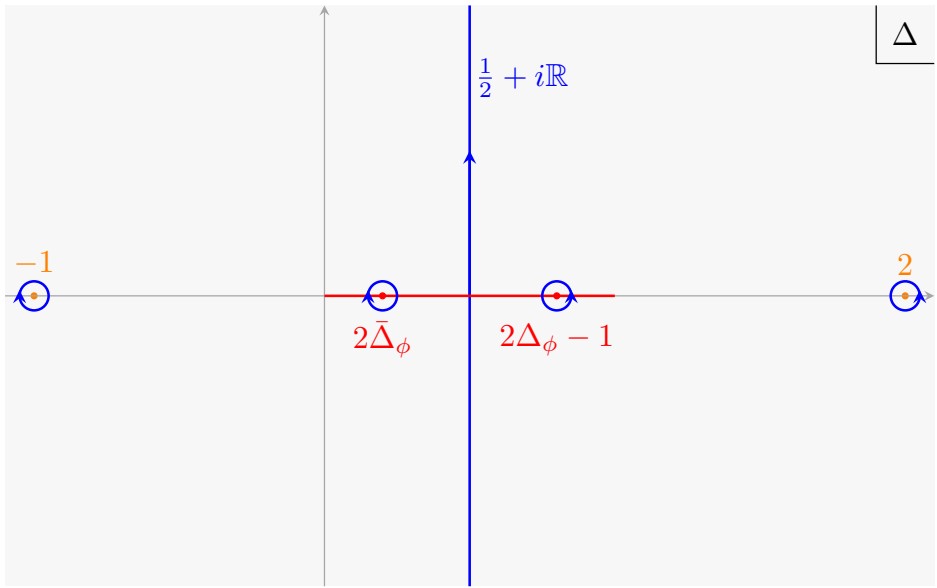

**Figure 5:** In blue, the contour of integration $\gamma$ in (4.12). A vertical line runs over the principal series and circles surround the poles corresponding to a discrete series and a complementary series UIR contributing to the spectral decomposition of the stress tensor. Because of shadow symmetry, the residues on a pole at $\Delta$ and $1 - \Delta$ are equal and opposite in sign. Here, we represented the case where the massive boson is in the complementary series and has $\Delta_\phi > 3/4$.

Finally, notice that in this special case of the free massive boson, the full spectral decomposition of the stress tensor can be expressed as one spectral integral with a modified contour. That is because there exists a regime of the parameters of the theory, namely in $d > 1$ and $\Delta_\phi \in \frac{d}{2} + i\mathbb{R} \cup (\frac{d}{4}, \frac{3d}{4})$, for which only the principal series contributes. We show this in appendix B.1. Then, since the two-point function of the stress tensor is an analytic function of $d$ and $\Delta_\phi$, the only thing that can happen is that poles in the principal series spectral density cross the integration contour and lead to extra contributions to the Källén-Lehmann decomposition as we continue in the mass of the scalar and in the dimensions. These poles can be accounted for by modifying the contour of integration, leading to the following decomposition in two dimensions

$$\langle T^{AB}(Y_1) T^{CD}(Y_2) \rangle = \int_\gamma \frac{d\Delta}{2\pi i} \frac{\varrho_\Theta^{\mathcal{P}}(\Delta)}{(\Delta + 1)^2 (\bar{\Delta} + 1)^2} \Pi_1^{AB} \Pi_2^{CD} G_\Delta(\sigma) , \qquad (4.12)$$

with the contour $\gamma$ shown in blue in figure 5.

### 4.2 Free massive fermion

As a second example, consider the theory of a free Majorana fermion in two dimensions

$$S = -\frac{1}{2} \int d^2 x \sqrt{g} \, \bar{\Psi} \left( \slashed{\nabla} + m \right) \Psi . \qquad (4.13)$$

This can be seen as a specific field parametrization of two-dimensional Ising field theory on the sphere above the critical temperature and with zero magnetic field. The Ising CFT

is reached when $m = 0$, and it notoriously has $c^{\text{UV}} = \frac{1}{2}$. The mass term $m\bar{\Psi}\Psi$ acts as a deforming operator which triggers a flow to the trivial empty theory in the IR, which has $c^{\text{IR}} = 0$.

We leave many details to appendix B.2. The canonically normalized two-point function is [48, 72]

$$\langle \Psi(x_1)\bar{\Psi}(x_2) \rangle = \frac{1}{\sqrt{\eta_1 \eta_2}} \begin{pmatrix} i[(\eta_1 + \eta_2) + (y_1 - y_2)]G_m^-(\sigma) & [(\eta_1 - \eta_2) + (y_1 - y_2)]G_m^+(\sigma) \\ [(y_1 - y_2) - (\eta_1 - \eta_2)]G_m^+(\sigma) & i[(\eta_1 + \eta_2) - (y_1 - y_2)]G_m^-(\sigma) \end{pmatrix}, \tag{4.14}$$

where

$$\begin{aligned}
G_m^+(\sigma) &\equiv \frac{1}{8}m \, \text{csch}(\pi m R) \, {}_2F_1\left(1 - imR, 1 + imR, 1, \frac{1+\sigma}{2}\right), \\
G_m^-(\sigma) &\equiv -\frac{i}{8}m^2 R \, \text{csch}(\pi m R) \, {}_2F_1\left(1 - imR, 1 + imR, 2, \frac{1+\sigma}{2}\right),
\end{aligned} \tag{4.15}$$

and we are working in flat slicing coordinates $ds^2 = R^2 \frac{-\mathrm{d}\eta^2 + \mathrm{d}y^2}{\eta^2}$ and $x^\mu = (\eta, y)$. The two-point invariant then takes the form

$$\sigma = \frac{\eta_1^2 + \eta_2^2 - (y_1 - y_2)^2}{2\eta_1 \eta_2}. \tag{4.16}$$

In appendix B.2 we show that, in the flat space limit, (4.14) reduces to the canonically normalized two-point function of a free fermion in two-dimensional flat space. The symmetric and conserved stress tensor for this theory is [73]

$$T_{\mu\nu} = \frac{1}{8}\bar{\Psi}\left(\Gamma_\mu \overset{\leftrightarrow}{\nabla}_\nu + \Gamma_\nu \overset{\leftrightarrow}{\nabla}_\mu\right)\Psi, \tag{4.17}$$

where $A\overset{\leftrightarrow}{\nabla}_\mu B \equiv A\left(\nabla_\mu B\right) - \left(\nabla_\mu A\right)B$, and $\Gamma_\mu$ are the Dirac gamma matrices in de Sitter, related to the flat space gamma matrices through the zweibein (see B.2 for an explanation). Using the equations of motion, the trace of the stress tensor reduces to

$$\Theta = -\frac{m}{2}\bar{\Psi}\Psi, \tag{4.18}$$

with two-point function

$$G_\Theta(\sigma) = \langle \Theta(x_1)\Theta(x_2) \rangle = 2m^2\left((1-\sigma)\left(G_m^+(\sigma)\right)^2 + (1+\sigma)\left(G_m^-(\sigma)\right)^2\right). \tag{4.19}$$

Applying our formula (2.30) we numerically verify that

$$c^{\text{UV}} - \lim_{R \to \infty} c_1(R) = \frac{1}{2}. \tag{4.20}$$

In contrast to the free boson case, there are no IR divergences associated with massless fermions in de Sitter, and $c_1(R)$ is well behaved throughout the flow. Using that $c^{\text{UV}} = \frac{1}{2}$, we show a numerical plot of $c_1(R)$ in figure 6. It is a monotonically decreasing function of the radius.

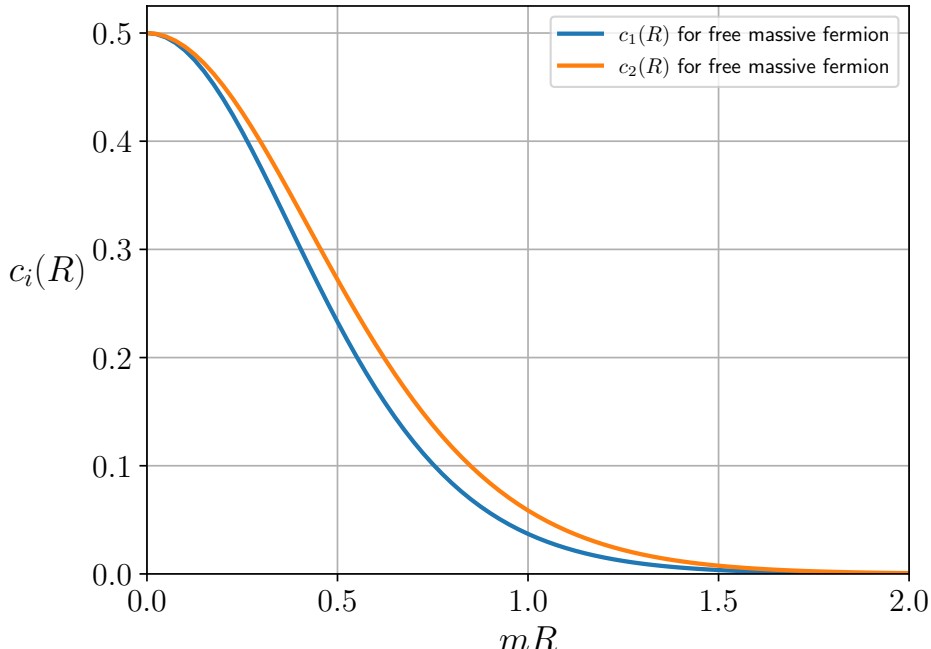

**Figure 6:** Numerical plots of $c_1(R)$ and $c_2(R)$ for the free massive fermion, obtained by using (2.30) and (3.19) with the two-point function of the trace of the stress tensor in this theory (4.19), and knowing that $c^{\text{UV}} = 1/2$. They interpolate between the critical Ising model in the UV and the empty theory in the IR.

To compute $c_2(R)$ for this theory, we start from the sum rule (3.19), which requires computing the integral of $r_2(\sigma)G_\Theta(\sigma)$ over the contour $\mathcal{C}_k$ shown in figure 3. We notice that (4.19) has a simple pole at $\sigma = 1$ and a branch cut at $\sigma \in [1, \infty)$. The sum rule thus becomes

$$c^{\text{UV}} - c_2(R) = \int_1^\infty d\sigma \; r_2(\sigma)\text{Disc}\left[R^4 G_\Theta(\sigma)\right] - 2\pi i \operatorname*{Res}_{\sigma=1}\left[r_2(\sigma)R^4 G_\Theta(\sigma)\right] \qquad (4.21)$$

The discontinuity can be computed analytically knowing that

$$\text{Disc}\left[\mathbf{F}(a, b, c, z)\right] = \frac{2\pi i}{\Gamma(a)\Gamma(b)}(z-1)^{c-a-b}\mathbf{F}\left(c-a, c-b, c-a-b+1, 1-z\right),$$

$$\text{Disc}\left[\mathbf{F}^2(a, b, c, z)\right] = \left(\text{Disc}[\mathbf{F}(a, b, c, z)] + 2\mathbf{F}(a, b, c, z)\text{Disc}[\mathbf{F}(a, b, c, z)]\right). \qquad (4.22)$$

The residue is simply

$$2\pi i \operatorname*{Res}_{\sigma=1}\left[r_2(\sigma)R^4 G_\Theta(\sigma)\right] = m^2 R^2 \left(2 - \frac{\pi^2}{3}\right). \qquad (4.23)$$

We evaluate the remaining integral numerically and plot the function $c_2(R)$ in figure 6. It is also a monotonic function of $R$, and it satisfies the condition (3.16).

### 4.3 A comment on the massless Schwinger model

The massless Schwinger model is an integrable QFT in $\text{dS}_2$ [74–76]. Here we show that its associated functions $c_1$ and $c_2$ are precisely the same as in the free massive scalar theory,

indicating that the two theories are related by a field redefinition as in flat space. The massless Schwinger model is defined through the following action

$$S = \int d^2x \sqrt{g} \left[ \bar{\Psi} \left( \slashed{\nabla} + i\slashed{A} \right) \Psi + \frac{1}{4q^2} F^{\mu\nu} F_{\mu\nu} \right] \tag{4.24}$$

where $\Psi$ is a Dirac spinor, $A_\mu$ is a compact $U(1)$ gauge field with field strength $F_{\mu\nu} = \partial_\mu A_\nu - \partial_\nu A_\mu$ and $q$ is the gauge coupling, which has mass dimensions 1. The trace of the stress tensor (2.5) is, on-shell,

$$\Theta = \frac{1}{2q^2} F^{\mu\nu} F_{\mu\nu} \tag{4.25}$$

The two-point function of $F$ in the Bunch-Davies vacuum is written explicitly in [74], and it has the precise form of the two-point function of a free massive boson (2.36):

$$\frac{1}{\sqrt{g_x g_y}} \langle F_{01}(x_1) F_{01}(x_2) \rangle = -\frac{q^4}{\pi} G_{\Delta_q}(\sigma) \,, \tag{4.26}$$

where in this case $\Delta_q(1 - \Delta_q) = \frac{1}{\pi} q^2 R^2$. Since in two dimensions this is the only degree of freedom of the field strength, this implies

$$G_\Theta(\sigma) = 2 \left( \frac{q^2}{\pi} \right)^2 \left( G_{\Delta_q}(\sigma) \right)^2 \,. \tag{4.27}$$

This is exactly the same two-point function as in (4.3). Since $c_1$ and $c_2$ can be derived through sum rules (2.30) and (3.19) which only depend on the trace of the stress tensor, they are precisely the same as for the free massive boson theory, up to the mapping $m^2 \leftrightarrow \frac{q^2}{\pi}$. This is not unexpected: it is well known in flat space that the massless Schwinger model can be mapped through a field redefinition to the free massive boson theory, precisely with $m^2 \leftrightarrow \frac{q^2}{\pi}$ [77, 78]. This equality of the c-functions hints to the fact that carefully bosonizing the action (4.24) should lead to the free massive scalar theory in dS$_2$ as well.

## 5 Discussion

In this work we have studied RG flows in unitary QFTs in dS$_2$ and $S^2$. We have introduced two functions of the radius which interpolate between the central charges of the CFTs that live at the fixed points of any RG flow. One is defined through certain components of the two-point function of the stress tensor at antipodal separation (2.29), while the other is the spectral weight of the traceless part of the stress tensor in the $\Delta = 2$ irrep (3.13). The fact that this spectral weight has to interpolate between the two central charges implies that it needs to be non-zero for any QFT, or in other words the stress tensor has to always couple to discrete series $\Delta = 2$ states. We have verified our formulas in the examples of the theories of a free massive boson, a free massive fermion and the Schwinger model. We showed that $c_2$ is monotonically decreasing in every case, while $c_1$ is monotonically decreasing in the free fermion flow and it is zero for all radii in the free boson case where the massless regime is ill-defined due to IR divergences. We found that the massless Schwinger model has the same $c_1$ and $c_2$ as the free boson theory. As an intermediate step, we worked out the

details of how the conservation of the stress tensor simplifies its spectral decomposition greatly. We argue that, in general, the sphere and de Sitter can be interesting background geometries to study QFT since the radius acts as a symmetry preserving IR regulator, and can be tuned to follow the RG flow and reveal new facts about QFTs of interest which may be inaccessible in flat space. Moreover, the existence and behavior of $c_1$ and $c_2$ are new rigorous constraints that any unitary QFT in $dS_2$ must satisfy.

There are some open questions which would be interesting to explore in the future:

- The c-functions we have introduced interpolate between $c^{UV}$ and $c^{IR}$. We also showed that $c^{UV} \geq c_i(R)$ for both, implying in particular Zamolodchikov's c-theorem $c^{UV} \geq c^{IR}$. In the two examples we studied, we also verified that they are monotonic for intermediate radii. It would be interesting to establish whether the monotonicity is true for all QFTs with a general proof or a counterexample.

- The examples in which we could test our formulas were gapped theories. In the future, we hope to test them in flows which have $c^{IR} \neq 0$, such as between minimal models in de Sitter.

- Can a similar approach to the one utilized in this work be adapted to the problem of finding RG-monotonic functions constructed from the stress tensor two-point function in higher dimensions? Analogously to what happens in AdS [56], the simple generalization of the differential equation (2.22) to higher dimensions is not enough to extract the trace anomalies, so a more sophisticated approach is required.

- The results of this paper can be thought of as a new set of constraints that any unitary QFT in $S^2/dS_2$ needs to satisfy. Some of them are in the form of positive sum rules on two-point functions of the stress tensor which relate IR and UV data. Combining these with constraints on higher-point functions one may be able to set up numerical bootstrap problems in de Sitter, as suggested in [44–47, 50, 79, 80].

- Are there any RG flows of interest for which the approach presented in this paper is more efficient than the well known flat space techniques? It would be interesting to understand whether there are computational advantages that come, for example, from the fact that one of the c-functions we propose is only dependent on the $\Delta = 2$ contribution to the spectral decomposition of the traceless part of the stress tensor, which has the form of a CFT two-point function of a spin 2 primary on the sphere.

## Acknowledgements

I am grateful to Dionysios Anninos, Tarek Anous, Victor Gorbenko, Grégoire Mathys, Joao Penedones, Jiaxin Qiao, Veronica Sacchi, Vladimir Schaub, Zimo Sun, Kamran Salehi Vaziri and Antoine Vuignier for insightful discussions. I am especially grateful to Joao Penedones for his consistent and supportive supervision, Vladimir Schaub for useful discussions about fermions in de Sitter and Grégoire Mathys for conversations on c-theorems and sum rules in flat space. Finally, I thank Dionysios Anninos and Tarek

Anous for hospitality at King's College and Queen Mary University in December 2023, a visit from which I learned a lot. I am supported by the Simons Foundation grant 488649 (Simons Collaboration on the Nonperturbative Bootstrap) and the Swiss National Science Foundation through the project 200020_197160 and through the National Centre of Competence in Research SwissMAP.

## A  Details on the position space sum rules

In this appendix we report some details concerning section (2.2) of the main text.

### A.1  Conservation equations and solutions of (2.22)

Here we report the three linearly independent constraints we get from imposing the conservation of the stress tensor on the functions $T_i(\sigma)$.

$$
\begin{aligned}
\frac{\sigma^2-1}{\sigma^3}\Big[ &-(d+4)\sigma^4(\sigma^2-1)T_1(\sigma) + (d^2+3d+4)\sigma^4 T_2(\sigma) + 4(\sigma+(d+1)\sigma^3)T_3(\sigma) \\
&+ (4+2d\sigma^2)T_5(\sigma) - \sigma^3(\sigma^2-1)^2 T_1'(\sigma) + (d+2)\sigma^3(\sigma^2-1)T_2'(\sigma) \\
&+ 4\sigma^2(\sigma^2-1)T_3'(\sigma) - (d+1)\sigma^3 T_4'(\sigma) - 2\sigma T_5'(\sigma) \Big] = 0\,, \\[4pt]
\frac{(\sigma^2-1)^2}{\sigma^3}\Big[ &(d+4)\sigma^4(\sigma^2-1)T_1(\sigma) - (d+4)\sigma^4 T_2(\sigma) - 2\sigma(2+(d+2)\sigma^2)T_3(\sigma) \\
&- 4T_5(\sigma) + \sigma^3(\sigma^2-1)^2 T_1'(\sigma) - 2\sigma^3(\sigma^2-1)T_2'(\sigma) - 4\sigma^2(\sigma^2-1)T_3'(\sigma) \\
&+ \sigma^3 T_4'(\sigma) + 2\sigma T_5'(\sigma) \Big] = 0\,, \\[4pt]
\frac{\sigma^2-1}{\sigma^4}\Big[ &(d+4)\sigma^4(\sigma^2-1)T_1(\sigma) - 2(d+2)\sigma^4 T_2(\sigma) - \sigma(4+2(d+2)\sigma^2+d(d+3)\sigma^4)T_3(\sigma) \\
&- (4+d\sigma^2)T_5(\sigma) + \sigma^3(\sigma^2-1)^2 T_1'(\sigma) - 2\sigma^3(\sigma^2-1)T_2'(\sigma) \\
&- \sigma^2(\sigma^2-1)(4+d\sigma^2)T_3'(\sigma) + \sigma^3 T_4'(\sigma) + \sigma(2+d\sigma^2)T_5'(\sigma) \Big] = 0\,.
\end{aligned}
\tag{A.1}
$$

They are obtained, respectively, by acting on $\nabla^A \langle T_{AB}(Y_1)T_{CD}(Y_2)\rangle = 0$ with the projectors

$$
V_1^B G_2^{CD}\,, \qquad V_1^B V_2^C V_2^D\,, \qquad G_{12}^{BC} V_2^D\,,
\tag{A.2}
$$

where the explicit form of these objects is in equation (2.8). In two dimensions, these are differential equations for the four $\mathcal{T}_i$ functions

$$
\begin{aligned}
&(2+\sigma^2)\mathcal{T}_1(\sigma) + \left(\sigma^2 + \frac{2}{\sigma^2-1}\right)\mathcal{T}_2(\sigma) - (\sigma^2+2)\mathcal{T}_3(\sigma) + \left(2+\sigma^2\right)\mathcal{T}_4(\sigma) \\
&+\sigma\left(\sigma^2-1\right)\mathcal{T}_1'(\sigma) + \sigma\left(1+2\sigma^2\right)\mathcal{T}_2'(\sigma) + \sigma\left(1-2\sigma^2\right)\mathcal{T}_3'(\sigma) - \sigma\mathcal{T}_4'(\sigma) = 0\,, \\[6pt]
&\qquad\quad -(\sigma^2+2)\mathcal{T}_1(\sigma) + \frac{\sigma^2-2}{\sigma^2-1}\mathcal{T}_2(\sigma) + 2\mathcal{T}_3(\sigma) - 2\mathcal{T}_4(\sigma) \\
&+\sigma(1-\sigma^2)\mathcal{T}_1'(\sigma) - \sigma(\sigma^2+1)\mathcal{T}_2'(\sigma) + \sigma(\sigma^2-1)\mathcal{T}_3'(\sigma) + \sigma\mathcal{T}_4'(\sigma) = 0\,, \\[6pt]
&-(2+\sigma^2)\mathcal{T}_1(\sigma) + \frac{\sigma^4-\sigma^2+2}{1-\sigma^2}\mathcal{T}_2(\sigma) + \frac{1}{2}(\sigma^2+4)\mathcal{T}_3(\sigma) - \frac{1}{2}(4+\sigma^2)\mathcal{T}_4(\sigma) \\
&+\sigma(1-\sigma^2)\mathcal{T}_1'(\sigma) - \sigma(\sigma^2+1)\mathcal{T}_2'(\sigma) + \frac{\sigma}{2}(\sigma^2-2)\mathcal{T}_3'(\sigma) + \frac{\sigma}{2}(2+\sigma^2)\mathcal{T}_4'(\sigma) = 0\,.
\end{aligned}
\tag{A.3}
$$

And here we report the kernels and functions $C$ that solve (2.22), parametrized by three real numbers $c_1$, $c_2$, $c_3$.

$$r(\sigma) = \frac{1}{2}\left[2c_3\sigma - 2c_2 + \sigma(c_1 - c_2)\log(1-\sigma) + \sigma(c_1 + c_2)\log(1+\sigma)\right],$$

$$
\begin{aligned}
C(\sigma) = \frac{1}{\sigma^2}\Big[ &-\left(1-\sigma^2\right)^2 \left(2(c_3 - c_1) + (c_1 - c_2)\log(1-\sigma) + (c_1 + c_2)\log(1+\sigma)\right)\mathcal{T}_1(\sigma) \\
&+ \left(2((\sigma^4 - 2\sigma^2 - 1)c_1 + c_3 + \sigma^2(c_3 + 2\sigma(c_2 - \sigma c_3)))\right) \\
&+ (1-\sigma^2)(2\sigma^2 + 1)(2\tanh^{-1}(\sigma)c_2 + c_1\log(1-\sigma^2)))\mathcal{T}_2(\sigma) \\
&+ (-2(\sigma^4 - 5\sigma^2 + 1)c_1 - \sigma(1 + 5\sigma^2)c_2 + 2(2\sigma^4 - 3\sigma^2 + 1)c_3 \\
&+ (2\sigma^4 - 3\sigma^2 + 1)(2\tanh^{-1}(\sigma)c_2 + c_1\log(1-\sigma^2)))\mathcal{T}_3(\sigma) \\
&+ ((2 - 4\sigma^2)c_1 + \sigma(\sigma^2 + 1)c_2 + 2(\sigma^2 - 1)c_3 \\
&+ (1-\sigma^2)((c_2 - c_1)\log(1-\sigma) - (c_1 + c_2)\log(\sigma + 1)))\mathcal{T}_4(\sigma)\Big]
\end{aligned}
\tag{A.4}
$$

The particular solution that leads to (2.30) is $c_1 = c_2 = -8\pi^2$ and $c_3 = 8\pi^2(\log(2) - 1)$.

## A.2  Coincident point limit of $T_i$

Here we derive the coincident point limit of the $T_i(\sigma)$ functions in (2.6). We use the following local coordinate system for de Sitter

$$Y^0 = \frac{1 - e^{-2t} + \mathbf{x}^2}{2e^{-t}}, \qquad Y^i = x^i e^t, \qquad Y^{d+1} = \frac{-1 - e^{-2t} + \mathbf{x}^2}{2e^{-t}}, \tag{A.5}$$

where $\mathbf{x} \in \mathbb{R}^d$ with $i = 1, \ldots, d$ and we keep $R = 1$. In this coordinate system, the metric is

$$ds^2 = -\mathrm{d}t^2 + e^{2t}\mathrm{d}\mathbf{x}^2. \tag{A.6}$$

The two-point invariant reads

$$\sigma = \frac{1}{2}e^{-(t_1+t_2)}\left(e^{2t_1} + e^{2t_2} + 2e^{2(t_1+t_2)}\mathbf{x}_1 \cdot \mathbf{x}_2 - e^{2(t_1+t_2)}(\mathbf{x}_1^2 + \mathbf{x}_2^2)\right). \tag{A.7}$$

Then, by using (2.3) we can compute the behavior of the tensor structures (2.7) near coincident points in these local coordinates, where $x^\mu = (t, \mathbf{x})$.

$$
\begin{aligned}
&\mathbb{T}_1^{\mu\nu\rho\sigma} \approx x^\mu x^\nu x^\rho x^\sigma, \qquad \mathbb{T}_2^{\mu\nu\rho\sigma} \approx \eta^{\mu\nu}x^\rho x^\sigma + x^\mu x^\nu \eta_{\rho\sigma} - x^\mu x^\nu x^\rho x^\sigma, \\
&\mathbb{T}_3^{\mu\nu\rho\sigma} \approx \eta^{\nu\sigma}x^\mu x^\rho + \eta^{\mu\sigma}x^\nu x^\rho + \eta^{\nu\rho}x^\mu x^\sigma + \eta^{\mu\rho}x^\nu x^\sigma, \\
&\mathbb{T}_4^{\mu\nu\rho\sigma} \approx \eta^{\rho\sigma}\eta^{\mu\nu} - \eta^{\rho\sigma}x^\mu x^\nu, \qquad \mathbb{T}_5^{\mu\nu\rho\sigma} \approx \eta^{\mu\sigma}\eta^{\nu\rho} + \eta^{\mu\rho}\eta^{\nu\sigma}.
\end{aligned}
\tag{A.8}
$$

This means the coincident point limit of our parametrization of the stress tensor (2.6) is

$$
\begin{aligned}
\langle T^{\mu\nu}(x)T^{\rho\sigma}(0)\rangle \approx &\; x^\mu x^\nu x^\rho x^\sigma T_1(x) + (\eta^{\mu\nu}x^\rho x^\sigma + \eta^{\rho\sigma}x^\mu x^\nu)T_2(x) \\
&+ (\eta^{\nu\sigma}x^\mu x^\rho + \eta^{\mu\sigma}x^\nu x^\rho + \eta^{\nu\rho}x^\mu x^\sigma + \eta^{\mu\rho}x^\nu x^\sigma)T_3(x) \\
&+ \eta^{\mu\nu}\eta^{\rho\sigma}T_4(x) + (\eta^{\mu\sigma}\eta^{\nu\rho} + \eta^{\mu\rho}\eta^{\nu\sigma})T_5(x).
\end{aligned}
\tag{A.9}
$$

We need to match with the well known flat space CFT two point function of the stress tensor (2.11). We can reshuffle the expression (2.11) in order to expand it in the same

tensor structures

$$\langle T^{\mu\nu}(x)T^{\rho\sigma}(0)\rangle_{\mathbb{M}}^{\text{CFT}} = \frac{4c_T}{x^{2d+6}}x^\mu x^\nu x^\rho x^\sigma - \frac{c_T}{x^{2d+4}}(\eta^{\nu\sigma}x^\mu x^\rho + \eta^{\mu\sigma}x^\nu x^\rho + \eta^{\nu\rho}x^\mu x^\sigma + \eta^{\mu\rho}x^\nu x^\sigma)$$
$$- \frac{c_T}{(d+1)x^{2d+2}}\eta^{\mu\nu}\eta^{\rho\sigma} + \frac{c_T}{2x^{2d+2}}(\eta^{\mu\sigma}\eta^{\nu\rho} + \eta^{\mu\rho}\eta^{\nu\sigma}) \tag{A.10}$$

By matching with our $T_i$ functions, we find the constraints mentioned in the main text

$$T_1 \approx \frac{4c_T}{x^{2d+6}}, \qquad T_2 \sim o(x^{-2d-2}), \qquad T_3 \approx -\frac{c_T}{x^{2d+4}},$$
$$T_4 \approx -\frac{c_T}{d+1}\frac{1}{x^{2d+2}}, \qquad T_5 \approx \frac{c_T}{2}\frac{1}{x^{2d+2}}. \tag{A.11}$$

## B  Details on the free scalar and the free fermion

Here we show some computational details and checks of our formulas in the cases of a free massive scalar and a free massive Majorana fermion.

### B.1  Free massive scalar

Consider the theory of a free massive scalar with $m^2R^2 = \Delta_\phi(d - \Delta_\phi)$. We work in $d+1$ dimensions, but we are ultimately interested in taking the limit $d \to 1$. We will thus ignore improvement terms in the stress tensor which arise from the conformal coupling in the action $\frac{d-1}{4d}R\phi^2$,[12] since in $d=1$ the coupling is zero.

$$S = -\frac{1}{2}\int d^{d+1}x\sqrt{|g|}\left(g^{\mu\nu}\partial_\mu\phi\partial_\nu\phi + m^2\phi^2\right). \tag{B.1}$$

As written in the main text, the stress tensor of this theory, which we here report uplifted to embedding space, is

$$T_{AB} = \nabla_A\phi\nabla_B\phi - \frac{1}{2}G_{AB}\left[\nabla^C\phi\nabla_C\phi + m^2\phi^2\right]. \tag{B.2}$$

We can split it into its traceless part and its trace. For convenience, we introduce some auxiliary vectors $W$ which are null ($W^2 = 0$) and tangent to the hypersurface in embedding space ($W\cdot Y = 0$), for the purpose of contracting all the indices while enforcing symmetricty and tracelessness [45, 48, 51, 52, 54, 72, 81]. Embedding space tensors are then traded for polynomials of $W$

$$\hat{T}(W) \equiv W^A W^B T_{AB} = (W\cdot\nabla)\phi(W\cdot\nabla)\phi. \tag{B.3}$$

To retrieve the expression of the traceless part of the stress tensor with indices, we act with the Todarov operator [45, 48, 72]

$$K_A \equiv \frac{d-1}{2}\left[\partial_{W^A} - Y_A(Y\cdot\partial_W)\right] + (W\cdot\partial_W)\partial_{W^A} - Y_A(Y\cdot\partial_W)(W\cdot\partial_W)$$
$$- \frac{1}{2}W_A\left[(\partial_W\cdot\partial_W) - (Y\cdot\partial_W)^2\right], \tag{B.4}$$

---

[12]In this expression R is the Ricci scalar.

in the following way

$$\hat{T}_{AB} = \frac{K_A K_B}{2(\frac{d-1}{2})_2} \hat{T}(W) \,. \tag{B.5}$$

Finally, let us mention that the covariant derivative has to be modified to accomodate the use of the $W$ vectors

$$\nabla_A = \partial_{Y^A} - Y_A (Y \cdot \partial_Y) - W_A (Y \cdot \partial_W) \,. \tag{B.6}$$

The first thing we do is to check whether there is some range of parameters for which the principal series is the only contribution to the Källén-Lehmann decomposition of $\hat{T}$, such that we can apply the inversion formulae from [45]. The criterion, also outlined in [45], is based on the fall-off of the components of the two-point function of $\hat{T}$ as we take $\sigma \to -\infty$. Let us write a generic two-point function of a spin 2 operator in index free formalism as

$$\langle \mathcal{O}(Y_1, W_1) \mathcal{O}(Y_2, W_2) \rangle = \sum_{m=0}^{2} (W_1 \cdot W_2)^{2-m} [(Y_1 \cdot W_2)(Y_2 \cdot W_1)]^m \mathcal{G}_m(\sigma) \,. \tag{B.7}$$

Then, the criterion for the principal series being the only class of UIRs appearing in the spectral decomposition of this two-point function is that the fall-offs of the $\mathcal{G}_m$ functions respect the following inequality

$$\lim_{\sigma \to -\infty} \mathcal{G}_m(\sigma) \sim |\sigma|^{-\omega_m - m} \,, \qquad \min_{m}[\mathrm{Re}(\omega_m)] > \frac{d}{2} + 2 \,, \tag{B.8}$$

When this condition is satisfied, the two-point function is square integrable when continued to EAdS, which ensures that harmonic functions in the principal series furnish a complete basis [81, 82], see section 4.3 in [45] for a detailed discussion. The two-point function of interest to us is

$$\langle \hat{T}(Y_1, W_1) \hat{T}(Y_2, W_2) \rangle = 2 \left[ (W_1 \cdot \nabla_1)(W_2 \cdot \nabla_2) \langle \phi(Y_1) \phi(Y_2) \rangle \right]^2 \,. \tag{B.9}$$

The fall-offs of its components in the basis (B.7) are

$$\min \omega_0 = \min \omega_1 = \min \omega_2 = 2 + 2\min(\mathrm{Re}\Delta_\phi, \mathrm{Re}\bar{\Delta}_\phi) \,. \tag{B.10}$$

We can thus say that for $\min(\mathrm{Re}\Delta_\phi, \mathrm{Re}\bar{\Delta}_\phi) > \frac{d}{4}$, the principal series is the only contribution to the spectral decomposition of $\hat{T}$. Let us start by assuming we are in this regime, which is satisfied when the free boson is in the principal series or in a portion of the complementary series $\Delta_\phi \in (\frac{d}{4}, \frac{3d}{4})$. Then, we can decompose the traceless part of the stress tensor in the principal series only

$$\langle \hat{T}(Y_1, W_1) \hat{T}(Y_2, W_2) \rangle = 2\pi \int_{\frac{d}{2} + i\mathbb{R}} \frac{d\Delta}{2\pi i} \left[ \varrho_{\hat{T},2}^{\mathcal{P}}(\Delta) G_{\Delta,2}(Y_1, Y_2; W_1, W_2) \right. $$
$$\left. + \varrho_{\hat{T},0}^{\mathcal{P}}(\Delta)(W_1 \cdot \nabla_1)^2 (W_2 \cdot \nabla_2)^2 G_\Delta(\sigma) \right] \,, \tag{B.11}$$

where we used the facts proven in section C.1 to exclude spin 1 contributions, and the explicit expression of $G_{\Delta,2}$ is given in (B.16). Applying the inversion formulae from

[45], specifically with the methods outlined in appendix H there, we compute the spectral densities in (B.11)

$$\varrho^{\mathcal{P}}_{\hat{T},2}(\Delta) = \frac{\lambda \sinh(\pi\lambda)\Gamma\left(\frac{2+\frac{d}{2}\pm i\lambda}{2}\right)^2}{2\pi^{3+\frac{d}{2}}\Gamma(\frac{d}{2}+2)\Gamma(2+\frac{d}{2}\pm i\lambda)} \prod_{\pm,\pm}\Gamma\left(\frac{2+\frac{d}{2}\pm i\lambda\pm 2i\lambda_\phi}{2}\right),\tag{B.12}$$

$$\varrho^{\mathcal{P}}_{\hat{T},0}(\Delta) = \frac{\left((d-1)\Delta\bar{\Delta}+4\Delta_\phi\bar{\Delta}_\phi\right)^2\lambda\sinh(\pi\lambda)\Gamma\left(\frac{\frac{d}{2}\pm i\lambda}{2}\right)^2}{2^8\pi^{3+\frac{d}{2}}d^2(\Delta+1)^2(\bar{\Delta}+1)^2\Gamma(\frac{d}{2})\Gamma(\frac{d}{2}\pm i\lambda)} \prod_{\pm,\pm}\Gamma\left(\frac{\frac{d}{2}\pm i\lambda\pm 2i\lambda_\phi}{2}\right),$$

where we are using $\Delta = \frac{d}{2}+i\lambda$ and $\Delta_\phi = \frac{d}{2}+i\lambda_\phi$ and the radius has been set to 1. The integral in (B.11) can then be checked numerically. We also independently compute $\varrho_{\Theta\hat{T}}$ and $\varrho_\Theta$ and we check that the identities (3.5) are verified. Using those identities and more in general what is discussed in section C.1, we can thus write the spectral decomposition of the full stress tensor two-point function for the free boson in the regime where $\min(\text{Re}\Delta_\phi, \text{Re}\bar{\Delta}_\phi) > \frac{d}{4}$:

$$\langle T^{AB}(Y_1)T^{CD}(Y_2)\rangle = 2\pi\int_{\frac{d}{2}+i\mathbb{R}}\frac{d\Delta}{2\pi i}\Big[\varrho^{\mathcal{P}}_{\hat{T},2}(\Delta)G^{AB,CD}_{\Delta,2}(Y_1,Y_2)$$
$$+\frac{\varrho^{\mathcal{P}}_\Theta(\Delta)}{d^2(\Delta+1)^2(\bar{\Delta}+1)^2}\Pi^{AB}_1\Pi^{CD}_2 G_\Delta(\sigma)\Big].\tag{B.13}$$

Now we start the continuation to $d = 1$. First of all, let us write the explicit form of $G_{\Delta,2}$, the free propagator of a massive traceless and transverse spin 2 field in de Sitter. In index-free notation, it is the solution to

$$\left(\nabla_1^2 - \Delta\bar{\Delta} - 2\right)G_{\Delta,2}(Y_1,Y_2;W_1,W_2) = 0, \qquad (K_1\cdot\nabla_1)G_{\Delta,2}(Y_1,Y_2;W_1,W_2) = 0,\tag{B.14}$$

with the extra condition of finiteness at antipodal separation. Because of $SO(1,d+1)$ invariance and the tangential condition $W_i\cdot Y_i = 0$, we can express the solution in terms of three scalar functions multiplying the elements of a polynomial of dot products involving the $W$ vectors

$$G_{\Delta,2}(Y_1,Y_2;W_1,W_2) = \sum_{m=0}^{2}(W_1\cdot W_2)^{2-m}[(W_1\cdot Y_2)(W_2\cdot Y_1)]^m\mathcal{G}_m(\sigma),\tag{B.15}$$

with [45]

$$\frac{\mathcal{G}_0(\sigma)}{N(\Delta)} = 8\left(2d(\mathbf{F}^{(0)}+\sigma\mathbf{F}^{(1)})+(\sigma^2 d-1)\mathbf{F}^{(2)}\right),\tag{B.16}$$

$$\frac{\mathcal{G}_1(\sigma)}{N(\Delta)} = 8\left(2d(d+1)\mathbf{F}^{(1)}+\sigma d(5+3d+\Delta\bar{\Delta})\mathbf{F}^{(2)}+(\sigma^2 d-1)(\Delta+2)(\bar{\Delta}+2)\mathbf{F}^{(3)}\right),$$

$$\frac{\mathcal{G}_2(\sigma)}{N(\Delta)} = 4(d)_3\mathbf{F}^{(2)}+(\Delta+2)(\bar{\Delta}+2)(4d(d+2)\sigma\mathbf{F}^{(3)}+(\sigma^2 d-1)(\Delta+3)(\bar{\Delta}+3)\mathbf{F}^{(4)}),$$

where we use a shorthand notation for some regularized hypergeometric functions

$$\mathbf{F}^{(a)} \equiv \mathbf{F}\left(\Delta + a, \bar{\Delta} + a, \frac{d+1}{2} + a, \frac{1+\sigma}{2}\right), \tag{B.17}$$

and here

$$N(\Delta) \equiv \frac{(\Delta + 1)(\bar{\Delta} + 1)\Gamma(\Delta)\Gamma(\bar{\Delta})}{2^{d+5}\pi^{\frac{d+1}{2}}d(\Delta - 1)(\bar{\Delta} - 1)}. \tag{B.18}$$

The index-open form of this propagator is then retrieved as

$$G_{\Delta,2}^{AB,CD}(Y_1, Y_2) = \frac{K_1^A K_1^B K_2^C K_2^D}{4\left(\frac{d-1}{2}\right)_2^2}G_{\Delta,2}(Y_1, Y_2; W_1, W_2). \tag{B.19}$$

Notably, the normalization factor (B.18) has simple poles at $\Delta = 1$ and $\bar{\Delta} = 1$, or equivalently at $\lambda = \pm i\frac{d-2}{2}$. When continuing in the number of dimensions, these poles will cross the integration contour over the principal series in (B.13) when passing by $d = 2$. The residues on their positions need to be added by hand in order to retrieve the correct Källén-Lehmann representation in $d = 1$. In [45], we showed that on these spurious poles, propagators and spectral densities associated to different spins are related to each other. The relations relevant here are

$$\operatorname*{Res}_{\Delta = d-1} G_{\Delta,2}(Y_1, Y_2; W_1, W_2) = \frac{2 - d}{d}(W_1 \cdot \nabla_1)^2(W_2 \cdot \nabla_2)^2 G_{d+1,0}(\sigma),$$
$$\varrho_{\hat{T},2}(d-1) = d(d-2)\operatorname*{Res}_{\Delta = d+1}\varrho_{\hat{T},0}(\Delta). \tag{B.20}$$

Using the conservation relations (3.5), we can further say

$$\varrho_{\hat{T},2}(d-1) = \frac{d-2}{d(d+2)^3}\left((d+2)\partial_\Delta\varrho_\Theta(d+1) - 2\varrho_\Theta(d+1)\right). \tag{B.21}$$

We thus see that in two dimensions ($d = 1$) there will be the appearance of a UIR with $\Delta = 2$ in the Källén-Lehmann representation of the traceless part of the stress tensor of a free massive boson. In particular, in this case $\varrho_\Theta(2) = 0$, and what we are left with is

$$\varrho_{\hat{T}}^{\mathcal{D}_2} = \frac{4\pi}{9}\partial_\Delta\varrho_\Theta^{\mathcal{P}}(2) = \frac{\lambda_\phi m^2}{3}\operatorname{csch}(2\pi\lambda_\phi), \tag{B.22}$$

where $m^2 = \frac{1}{4} + \lambda_\phi^2$ ($R = 1$ here). Finally, in $d = 1$, the following identities are true

$$\left(Y_1 \cdot W_2^\pm\right)\left(Y_2 \cdot W_1^\pm\right) = (\sigma + 1)\left(W_1^\pm \cdot W_2^\pm\right),$$
$$\left(Y_1 \cdot W_2^\mp\right)\left(Y_2 \cdot W_1^\pm\right) = (\sigma - 1)\left(W_1^\pm \cdot W_2^\mp\right). \tag{B.23}$$

where $\pm$ stands for the $SO(1,2)$ chirality. These identities stem from the fact that spin $J$ tensors have only two independent components in two dimensions, corresponding to two $SO(1,2)$-inequivalent $W^A$. Every two-point function of spin $J$ operators in two dimensions can be then decomposed in two components, one proportional to $(W_1^\pm \cdot W_2^\pm)^J$ and one proportional to $(W_1^\pm \cdot W_2^\mp)^J$. The second one is vanishing except if the theory violates parity.

It can be checked that, using (B.23) in (B.16), both components of the two-point function $G_{\Delta,2}$ vanish in two dimensions. All in all, the spectral decomposition of the stress tensor of a free massive boson in the principal series in two dimensions, obtained by continuing in $d$ from (B.13), is

$$\langle T^{AB}(Y_1)T^{CD}(Y_2)\rangle = 2\pi \int_{\frac{1}{2}+i\mathbb{R}} \frac{d\Delta}{2\pi i} \frac{\varrho_\Theta^{\mathcal{P}}(\Delta)}{(\Delta+1)^2(\bar{\Delta}+1)^2} \Pi_1^{AB}\Pi_2^{CD} G_\Delta(\sigma)$$
$$+ \varrho_{\hat{T}}^{\mathcal{D}_2} \Pi_1^{AB}\Pi_2^{CD} G_{\Delta=2}(\sigma)\,, \tag{B.24}$$

with the spectral densities given by setting $d=1$ in (B.12) and (B.22).

**Complementary series contributions**  Until now we had assumed the free scalar sits in the range $\min(\mathrm{Re}\Delta_\phi, \mathrm{Re}\bar{\Delta}_\phi) > \frac{1}{4}$, or equivalently $m^2 > 3/16$. We can analytically continue (B.24) beyond that regime. From the explicit expression of $\varrho_\Theta(\Delta)$ in (B.12) we see that poles at $\lambda = \pm 2\lambda_\phi + \frac{i}{2}$ cross the integration contour over the principal series when $|\mathrm{Im}\lambda_\phi| > \frac{1}{4}$. Summing the residues on these poles, we obtain the full decomposition

$$\langle T^{AB}(Y_1)T^{CD}(Y_2)\rangle = 2\pi \int_{\frac{1}{2}+i\mathbb{R}} \frac{d\Delta}{2\pi i} \frac{\varrho_\Theta^{\mathcal{P}}(\Delta)}{(\Delta+1)^2(\bar{\Delta}+1)^2} \Pi_1^{AB}\Pi_2^{CD} G_\Delta(\sigma)$$
$$+ \theta\left(\mathrm{Re}\Delta_\phi - \frac{3}{4}\right)\int_0^1 d\Delta \frac{\varrho_\Theta^{\mathcal{C}}(\Delta)}{(\Delta+1)^2(\bar{\Delta}+1)^2} \Pi_1^{AB}\Pi_2^{CD} G_\Delta(\sigma) + (\Delta_\phi \to 1-\Delta_\phi)$$
$$+ \varrho_{\hat{T}}^{\mathcal{D}_2} \Pi_1^{AB}\Pi_2^{CD} G_{\Delta=2}(\sigma)\,, \tag{B.25}$$

with

$$\varrho_\Theta^{\mathcal{C}}(\Delta) = -\delta(\Delta - 2\Delta_\phi + 1)\frac{(\Delta+1)^2\bar{\Delta}\cos(\pi\Delta)\Gamma(\frac{3}{2}-\Delta)\Gamma(\frac{3-\Delta}{2})\Gamma(\frac{\Delta}{2})^2}{2^{4-\Delta}\pi^2\Gamma(1-\frac{\Delta}{2})}\,, \tag{B.26}$$

and where $\theta(x)$ is a Heaviside theta function.

Notice that all these extra terms can be added as a modification of the original contour of integration, becoming

$$\langle T^{AB}(Y_1)T^{CD}(Y_2)\rangle = 2\pi \int_\gamma \frac{d\Delta}{2\pi i} \frac{\varrho_\Theta^{\mathcal{P}}(\Delta)}{(\Delta+1)^2(\bar{\Delta}+1)^2} \Pi_1^{AB}\Pi_2^{CD} G_\Delta(\sigma)\,, \tag{B.27}$$

where $\gamma$ is the contour represented in blue in figure 5.

**The c-functions**  By using the techniques outlined in appendix C.2, we compute the function $c_1(R)$ from its definition (2.29). For the free boson in two-dimensions we obtain, in particular

$$\mathcal{T}_3(-1) = \frac{m^4}{128}\csc^2(\pi\Delta_\phi)\,, \qquad \mathcal{T}_4(-1) = 3\frac{m^4}{128}\csc^2(\pi\Delta_\phi)\,. \tag{B.28}$$

Using (2.29) we thus get $c_1(R) = 0$, which is due to the IR divergences associated to massless scalars in de Sitter, affecting the trace of the stress tensor. The second c-function is instead only dependent on the traceless part, and we reported its explicit expression in the main text (4.11).

## B.2 Free massive fermion

Consider the theory of a free massive Majorana fermion in two-dimensional de Sitter space, described by the action

$$S = -\frac{1}{2} \int d^2x \sqrt{|g|}\, \bar{\Psi} \left( \slashed{\nabla} + m \right) \Psi \,. \tag{B.29}$$

The only spin $\frac{1}{2}$ UIRs are in the principal series, with mass and conformal weight related through $\Delta = \frac{1}{2} + imR$, with $m > 0$ [83, 84]. We choose to work with conventions in which $\Psi$ is a real bispinor

$$\Psi = \begin{pmatrix} \psi_1 \\ \psi_2 \end{pmatrix}, \tag{B.30}$$

where $\psi_1$ and $\psi_2$ are real Grassmann functions. Moreover, it is useful to go to local coordinates, and we choose the flat slicing metric $ds^2 = R^2 \frac{-d\eta^2 + dy^2}{\eta^2}$. Then, we choose the (flat) gamma matrices to be

$$\gamma_0 = \begin{pmatrix} 0 & 1 \\ -1 & 0 \end{pmatrix}, \qquad \gamma_1 = \begin{pmatrix} 0 & 1 \\ 1 & 0 \end{pmatrix}. \tag{B.31}$$

The corresponding gamma matrices in de Sitter are given by $\Gamma^\mu = e^\mu_a \gamma^a$, with the zweibein satisfying $e^a_\mu e^b_\nu \eta_{ab} = g_{\mu\nu}$. With these conventions, the charge conjugation matrix, defined by

$$C\gamma_\mu C^{-1} = -\gamma_\mu^T \,, \tag{B.32}$$

can be chosen to be $C = \begin{pmatrix} 0 & 1 \\ -1 & 0 \end{pmatrix}$. Then, we have that $\bar{\Psi} = \begin{pmatrix} -\psi_2 & \psi_1 \end{pmatrix}$, and the two-point function is

$$\langle \Psi(x_1)\bar{\Psi}(x_2)\rangle = \begin{pmatrix} -\langle\psi_1(x_1)\psi_2(x_2)\rangle & \langle\psi_1(x_1)\psi_1(x_2)\rangle \\ -\langle\psi_2(x_1)\psi_2(x_2)\rangle & \langle\psi_2(x_1)\psi_1(x_2)\rangle \end{pmatrix}. \tag{B.33}$$

As explained in the main text, the trace of the stress tensor in this theory is

$$\Theta(x) = -\frac{m}{2}\bar{\Psi}\Psi(x) = -m\psi_1\psi_2(x)\,, \tag{B.34}$$

and the associated two-point function is

$$\langle\Theta(x_1)\Theta(x_2)\rangle = m^2 (\langle\psi_1(x_1)\psi_2(x_2)\rangle\langle\psi_2(x_1)\psi_1(x_2)\rangle - \langle\psi_1(x_1)\psi_1(x_2)\rangle\langle\psi_2(x_1)\psi_2(x_2)\rangle)\,. \tag{B.35}$$

The entries of the matrix (B.33) that solve the equations of motion [48, 85]

$$\left( \slashed{\nabla} + m \right) \Psi = 0 \qquad \longrightarrow \qquad \left( \eta\gamma^\mu \partial_\mu + \frac{1}{2}\gamma_0 + m \right) \Psi = 0\,, \tag{B.36}$$

were given in eq. (4.14) [48, 72].

We are now going to show that, in the flat space limit, we reproduce the correct two-point function, thus providing an independent check of the normalization presented in the references [48, 72].

**Flat space limit**   Let us focus on

$$\mathcal{G}^-(\sigma) \equiv -\langle \psi_1(x_1)\psi_2(x_2)\rangle = \frac{i[(\eta_1 + \eta_2) + (y_1 - y_2)]}{\sqrt{\eta_1 \eta_2}} G_m^-(\sigma)\,,$$

$$\mathcal{G}^+(\sigma) \equiv \langle \psi_1(x_1)\psi_1(x_2)\rangle = \frac{[(\eta_1 - \eta_2) + (y_1 - y_2)]}{\sqrt{\eta_1 \eta_2}} G_m^+(\sigma)\,,$$

(B.37)

with $G_m^+$ and $G_m^-$ given in (4.15).

As usual, we start by taking $\eta \to t - R$ and $y \to x$. Then

$$\sigma = \frac{\eta_1^2 + \eta_2^2 - (y_1 - y_2)^2}{2\eta_1\eta_2} \to 1 - \frac{-(t_1 - t_2)^2 + (x_1 - x_2)^2}{2R^2} \equiv 1 - \frac{x^2}{2R^2}\,.$$

(B.38)

After some simplifications, we obtain

$$\mathcal{G}^-(\sigma) \to -\frac{m^2 R}{4}\, \mathrm{csch}(\pi m R)\, {}_2F_1\left(1 - imR, 1 + imR, 2, 1 - \frac{x^2}{4R^2}\right)\,,$$

$$\mathcal{G}^+(\sigma) \to \frac{m\,\mathrm{csch}(\pi m R)}{8R}(t_1 - t_2 + x_1 - x_2)\, {}_2F_1\left(1 - imR, 1 + imR, 1, 1 - \frac{x^2}{4R^2}\right)\,.$$

(B.39)

We use the following Barnes representation of the regularized hypergeometric function

$$\mathbf{F}(a, b, c, z) = \frac{\int_{\mathbb{R}+i\epsilon} ds\, \Gamma(a + is)\Gamma(b + is)\Gamma(c - a - b - is)\Gamma(-is)(1 - z)^{is}}{2\pi\Gamma(a)\Gamma(b)\Gamma(c - a)\Gamma(c - b)}\,.$$

(B.40)

We can apply it directly to $\mathcal{G}^-$ without any issues. Using $\Gamma(a \pm ib) \equiv \Gamma(a + ib)\Gamma(a - ib)$, we write

$$\mathcal{G}^-(\sigma) \to -\frac{m^2 R\, \mathrm{csch}(\pi m R)}{8\pi\Gamma(1 \pm imR)^2} \int_{\mathbb{R}+i\epsilon} ds\, \Gamma(1 \pm imR + is)\Gamma(-is)^2 \left(\frac{x_{12}^2}{4R^2}\right)^{is}$$

(B.41)

For $\mathcal{G}^+$, instead, there is a subtlety: the contour in (B.40) does not actually separate the two series of poles in the gamma functions. We thus need to introduce a regulator which we take to be $\alpha \geq 1$ and we eventually will take to 0, and write

$$\mathcal{G}^+_{(\alpha)}(\sigma) \equiv \frac{m\,\mathrm{csch}(\pi m R)}{8R}(t_1 - t_2 + x_1 - x_2)\mathbf{F}\left(1 - imR, 1 + imR, 1 + \alpha, 1 - \frac{x^2}{4R^2}\right)\,.$$

(B.42)

Then, the Barnes representation for the regulated $\mathcal{G}^+_{(\alpha)}(\sigma)$ reads

$$\mathcal{G}^+_{(\alpha)}(\sigma) \to \frac{m\,\mathrm{csch}(\pi m R)(t_1 - t_2 + x_1 - x_2)}{16\pi R\Gamma(\pm imR)\Gamma(1 \pm imR)}$$

$$\times \int_{\mathbb{R}+i\epsilon} ds\, \Gamma(1 \pm imR + is)\Gamma(-is)\Gamma(\alpha - 1 - is)\left(\frac{x^2}{4R^2}\right)^{is}$$

(B.43)

where we already took the regulator to zero where it didn't cause problems. Now we take the large radius limit. In this limit,

$$\Gamma(a \pm ibR) \to 2\pi e^{-\pi bR}(bR)^{2a-1}\,,$$

(B.44)

and only the growing part of $\operatorname{csch}(\pi m R)$ matters

$$\mathcal{G}^-(\sigma) \to -\frac{m}{8\pi^2} \int_{\mathbb{R}+i\epsilon} ds \left(\frac{m^2 x^2}{4}\right)^{is} \Gamma(-is)^2\,,$$
$$\mathcal{G}^+_{(\alpha)}(\sigma) \to \frac{m^2}{16\pi^2}((t_1 - t_2) + (x_1 - x_2)) \int_{\mathbb{R}+i\epsilon} ds \left(\frac{m^2 x^2}{4}\right)^{is} \Gamma(\alpha - 1 - is)\Gamma(-is)\,. \tag{B.45}$$

Here we recognize the Barnes representations of the modified Bessel function of the second kind

$$K_\nu(z) = \frac{1}{4\pi i} \left(\frac{z}{2}\right)^\nu \int_{c+i\mathbb{R}} dt\, \Gamma(t)\Gamma(t-\nu) \left(\frac{z}{2}\right)^{-2t}\,, \tag{B.46}$$

with $c > \max(\operatorname{Re}(\nu), 0)$. For $\mathcal{G}_-$, the result is spot on. For $\mathcal{G}^{(\alpha)}_+$, the validity of the integral representation (B.46) depends on $\alpha$, specifically $\alpha \geq 1$, which is precisely the values for which (B.43) is valid. We can thus substitute also here the Bessel function, and we obtain

$$\mathcal{G}^-(\sigma) \to -\frac{m}{2\pi} K_0(m|x|)\,,$$
$$\mathcal{G}^+_{(\alpha)}(\sigma) \to \frac{m^2}{4\pi} \left(\frac{2}{m|x|}\right)^{1-\alpha} (t_1 - t_2 + x_1 - x_2) K_{1-\alpha}(m|x|)\,. \tag{B.47}$$

The Bessel function is analytic in its order. We can thus now continue to $\alpha = 0$ and obtain

$$\mathcal{G}^-(\sigma) \to -\frac{m}{2\pi} K_0(m|x|)\,,$$
$$\mathcal{G}^+(\sigma) \to \frac{m}{2\pi} \frac{(t_1 - t_2 + x_1 - x_2)}{|x|} K_1(m|x|)\,. \tag{B.48}$$

Summarizing, we have shown that in the flat space limit

$$\langle \Psi(x_1)\bar{\Psi}(x_2) \rangle \to \frac{m}{2\pi} \begin{pmatrix} -K_0(m|x|) & \frac{t_1+x_1-t_2-x_2}{|x|} K_1(m|x|) \\ \frac{t_1-x_1-t_2+x_2}{|x|} K_1(m|x|) & -K_0(m|x|) \end{pmatrix}\,, \tag{B.49}$$

precisely matching the canonical normalization (see for example [86]).

# C   Details on the spectral decomposition of the stress tensor

In this section we provide extra details regarding the spectral decomposition of the stress tensor. First, we prove the relations (3.5) used extensively in the main text. Then, we relate the $T_i(\sigma)$ and the $\mathcal{T}_i(\sigma)$ defined in (2.6) and (2.18) to integrals over spectral densities. This in turn allows us to prove that these functions are always finite at $\sigma = -1$, a fact that is crucial in deriving the form of $c_1(R)$ in (2.29).

## C.1   General relations among the spectral densities

Here we prove the relations (3.5) used in the main text. We start from the most general spectral decomposition of a spin 2 symmetric tensor in $\mathrm{dS}_{d+1}/S^{d+1}$, for which we already

explained the notation in section 3

$$\langle T^{AB}(Y_1)T^{CD}(Y_2)\rangle = 2\pi \int_{\frac{d}{2}+i\mathbb{R}} \frac{d\Delta}{2\pi i}\left[ \sum_{\ell=0}^{2} \varrho_{\hat{T},\ell}^{\mathcal{P}}(\Delta)G_{\Delta,\ell}^{AB,CD}(Y_1,Y_2) \right.$$
$$+ \varrho_{\hat{T}\Theta}^{\mathcal{P}}(\Delta)\left( \frac{G_2^{CD}}{d+1}\hat{\Pi}_1^{AB}G_\Delta(\sigma) + \frac{G_1^{AB}}{d+1}\hat{\Pi}_2^{CD}G_\Delta(\sigma)\right)$$
$$\left. + \varrho_\Theta^{\mathcal{P}}(\Delta)\frac{G_1^{AB}G_2^{CD}}{(d+1)^2}G_\Delta(\sigma)\right] + \text{other UIRs}\,, \qquad \text{(C.1)}$$

where the other UIRs can be complementary series, which has the same analytic expression as the principal series but is integrated over $\Delta \in (0,1)$, exceptional series type I and exceptional series type II [45, 67]. We impose conservation, so from now on we consider the equation

$$\nabla_A\langle T^{AB}(Y_1)T^{CD}(Y_2)\rangle = 0\,. \qquad \text{(C.2)}$$

Various manipulations of this equation will lead to the spectral relations (3.5). The complementary series contributions have the same functional form as the principal series ones, so every relation we are going to find is valid also for the spectral densities on the complementary series. We thus drop the superscript $\mathcal{P}$ and consider a generic contour for the integral over $\Delta$. We now start to consider the consequences of (C.2).

First of all, the divergence kills the $\ell = 2$ term in the sum, which is automatically and independently conserved by definition. The first nontrivial statement comes from taking a trace over the indices $C, D$

$$\int d\Delta \Big[ \varrho_{\hat{T}\Theta}(\Delta)\nabla_{1A}\hat{\Pi}^{AB} + \frac{\varrho_\Theta(\Delta)}{d+1}\nabla_{1A}G_1^{AB}\Big]G_\Delta(\sigma) = 0 \qquad \text{(C.3)}$$

Using the explicit expressions of the projectors (3.2), the induced metric and the covariant derivative (2.4), we find that (C.3) implies

$$\varrho_{\hat{T}\Theta}(\Delta) = \frac{\varrho_\Theta(\Delta)}{d(\Delta+1)(\bar{\Delta}+1)}\,. \qquad \text{(C.4)}$$

Now we use this fact, and (C.2) becomes

$$\int d\Delta \Big[ \varrho_{\hat{T},1}(\Delta)\nabla_{1A}G_{\Delta,1}^{AB,CD}(Y_1,Y_2) + \varrho_{\hat{T},0}(\Delta)\nabla_{1A}\hat{\Pi}_1^{AB}\hat{\Pi}_2^{CD}G_\Delta(\sigma)$$
$$+ \varrho_\Theta(\Delta)\Big(\frac{1}{d(\Delta+1)(\bar{\Delta}+1)}\Big(\nabla_{1A}\hat{\Pi}_1^{AB}\frac{G_2^{CD}}{d+1} + \nabla_{1A}\frac{G_1^{AB}}{d+1}\hat{\Pi}_2^{CD}\Big) \qquad \text{(C.5)}$$
$$+ \nabla_{1A}\frac{G_1^{AB}G_2^{CD}}{(d+1)^2}\Big)G_\Delta(\sigma)\Big] = 0\,,$$

where we used that by definition $G_{\Delta,0}^{AB,CD} = \hat{\Pi}_1^{AB}\hat{\Pi}_2^{CD}G_\Delta$. Now, carrying out all the necessary computations, we find the last two relations

$$\varrho_{\hat{T},0}(\Delta) = \frac{\varrho_\Theta(\Delta)}{d^2(\Delta+1)^2(\bar{\Delta}+1)^2}\,, \qquad \varrho_{\hat{T},1}(\Delta) = 0\,, \qquad \text{(C.6)}$$

where we used the fact that the term proportional to $\varrho_{\hat{T},0}$ turns out to have the same tensor structure as the one proportional to $\varrho_\Theta$, while $\varrho_{\hat{T},1}$ has an independent tensor structure, and thus has to vanish on its own. This shows a fact that is well known in flat space: the stress tensor cannot interpolate between the vacuum and states carrying $SO(d)$ spin 1.

Using all of these relations, the expressions simplify greatly, and all the terms associated to $\varrho_\Theta$, $\varrho_{\hat{T}\Theta}$ and $\varrho_{\hat{T},0}$ collapse into one single term, resulting in

$$\langle T^{AB}(Y_1)T^{CD}(Y_2)\rangle =2\pi \int_{\frac{d}{2}+i\mathbb{R}} \frac{d\Delta}{2\pi i}\Big[\varrho^{\mathcal{P}}_{\hat{T},2}(\Delta)G^{AB,CD}_{\Delta,2}(Y_1,Y_2)$$
$$+\frac{\varrho^{\mathcal{P}}_\Theta(\Delta)}{d^2(\Delta+1)^2(\bar{\Delta}+1)^2}\Pi_1^{AB}\Pi_2^{CD}G_\Delta(\sigma)\Big] \tag{C.7}$$
$$+ \text{other UIRs}\,,$$

with $\Pi_i^{AB}$ defined in (3.7).

In **two dimensions**, the argument is very similar. The most general decomposition of a spin 2 symmetric tensor is

$$\langle T^{AB}(Y_1)T^{CD}(Y_2)\rangle =2\pi \int_{\frac{1}{2}+i\mathbb{R}} \frac{d\Delta}{2\pi i}\Bigg[\varrho^{\mathcal{P}}_{\hat{T},1}(\Delta)G^{AB,CD}_{\Delta,1}(Y_1,Y_1)+\varrho^{\mathcal{P}}_{\hat{T},0}(\Delta)\hat{\Pi}_1^{AB}\hat{\Pi}_2^{CD}G_\Delta(\sigma)$$
$$+\varrho^{\mathcal{P}}_{\hat{T}\Theta}(\Delta)\left(\hat{\Pi}_1^{AB}\frac{G_2^{CD}}{2}G_\Delta(\sigma)+\frac{G_1^{AB}}{2}\hat{\Pi}_2^{CD}G_\Delta(\sigma)\right)$$
$$+\frac{G_1^{AB}G_2^{CD}}{4}\varrho^{\mathcal{P}}_\Theta(\Delta)G_\Delta(\sigma)\Bigg]+\text{complementary series} \tag{C.8}$$
$$+\varrho^{\mathcal{D}_1}_{\hat{T}}\hat{\Pi}_1^{AB}\hat{\Pi}_2^{CD}G_{\Delta=1}(\sigma)+\varrho^{\mathcal{D}_2}_{\hat{T}}\hat{\Pi}_1^{AB}\hat{\Pi}_2^{CD}G_{\Delta=2}(\sigma)$$

where, group theoretically, the first two terms stand for the contributions from states carrying the two inequivalent chiralities of $SO(1,2)$ (see [45] for an in-depth discussion on this), and of course states do not carry any spin in two dimensions, so there is no $\ell=2$ contribution. The only difference between the complementary and principal series contributions will be again the domain of integration, while the discrete series has been explicitly added. Contributions from this series of UIRs can only be traceless because the trace of the stress tensor is a scalar operator, and as such it cannot carry discrete series irreps. More in general, local operators with spin $J$ can only couple to discrete series states with $\Delta \leq J$ [45, 50].

Now we impose conservation (C.2). For the principal and complementary series contributions, the computations are analogous to what was done in the previous paragraph, and so one obtains (C.4) and (C.6) with $d=1$. For the discrete series, something interesting happens: we have that

$$\nabla_{1A}\hat{\Pi}_1^{AB}\hat{\Pi}_2^{CD}G_{\Delta=2}(\sigma) = 0\,, \qquad \nabla_{1A}\hat{\Pi}_1^{AB}\hat{\Pi}_2^{CD}G_{\Delta=1}(\sigma) \neq 0\,. \tag{C.9}$$

This implies that necessarily $\varrho^{\mathcal{D}_1}_{\hat{T}}(\Delta)=0$, or in other words the stress tensor cannot interpolate between the vacuum and states in the $\Delta=1$ discrete series irrep. Instead, the $\Delta=2$

irrep is allowed and is conserved independently from the principal and complementary series contributions. Using all these facts together, we can write

$$
\begin{aligned}
\langle T^{AB}(Y_1)T^{CD}(Y_2)\rangle =& 2\pi \int_{\frac{1}{2}+i\mathbb{R}} \frac{d\Delta}{2\pi i}\, \frac{\varrho_\Theta^{\mathcal{P}}(\Delta)}{(\Delta+1)^2(\bar{\Delta}+1)^2}\Pi_1^{AB}\Pi_2^{CD}G_\Delta(\sigma) \\
& + \int_0^1 d\Delta\, \frac{\varrho_\Theta^{\mathcal{C}}(\Delta)}{(\Delta+1)^2(\bar{\Delta}+1)^2}\Pi_1^{AB}\Pi_2^{CD}G_\Delta(\sigma) \\
& + \varrho_{\hat{T}}^{\mathcal{D}_2}\Pi_1^{AB}\Pi_2^{CD}G_{\Delta=2}(\sigma)\,,
\end{aligned}
\tag{C.10}
$$

which is the complete and general spectral decomposition of the stress tensor two-point function in $S^2/\mathrm{dS}_2$. In section 3 we have shown that in any unitary QFT $\varrho_{\hat{T}}^{\mathcal{D}_2}$ interpolates between $c^{\mathrm{UV}}$ and $c^{\mathrm{IR}}$. This implies that the $\Delta=2$ contribution to the spectral decomposition of the stress tensor is not just *allowed* but rather it is *necessary* in any unitary QFT in $S^2/\mathrm{dS}_2$.

## C.2 How to compute the $T_i$ functions

Throughout this work, we have used two decompositions of the stress tensor, namely the ones in terms of tensor structures (2.6) and the ones in terms of the spectral densities (3.6), (3.8). Here, we are going to show relations between the two, which are crucial in deriving formulas for $c^{\mathrm{UV}}$ and $c_1(R)$ independently of their difference. The main idea is to look closely at the explicit expressions of the tensor structures $\mathbb{T}_i$ given in (2.7). We notice that there are some combinations of the coordinates and the metric with specific indices that appear uniquely in each tensor structure. Specifically,

$$
\begin{aligned}
\mathbb{T}_5^{ABCD} &\overset{!}{\supset} \{\eta^{AD}\eta^{BC},\eta^{AC}\eta^{BD}\}\,, \\
\mathbb{T}_4^{ABCD} &\overset{!}{\supset} \{\eta^{AB}\eta^{CD}\}\,, \\
\mathbb{T}_3^{ABCD} &\overset{!}{\supset} \{\sigma\eta^{BD}Y_1^A Y_1^C,\sigma\eta^{AD}Y_1^B Y_1^C\}\,, \\
\mathbb{T}_2^{ABCD} &\overset{!}{\supset} \{\eta^{AB}Y_1^C Y_1^D,-\sigma\eta^{CD}Y_1^B Y_2^A,-\sigma\eta^{CD}Y_1^A Y_2^B,\sigma\eta^{CD}Y_2^A Y_2^B\}\,.
\end{aligned}
\tag{C.11}
$$

whre with the symbol $\overset{!}{\supset}$ we mean the terms on the right hand side appear only in the tensor structure on the left hand side and not in the others.

Finding the coefficients of any of the terms on the right hand side thus uniquely identifies the $T_i$ component within a two-point function. For $T_1$, it is sufficient to subtract all other contributions. Here, we will do this in (3.6) in order to find relations between the

$T_i$ and integrals of spectral densities. We obtain

$$T_5(\sigma) = 2\pi \int_{\frac{d}{2}+i\mathbb{R}} \frac{d\Delta}{2\pi i} \left( \frac{1}{2} \varrho^{\mathcal{P}}_{\hat{T},2}(\Delta)\mathcal{G}_0(\sigma) + \varrho^{\mathcal{P}}_{\hat{T},0}(\Delta)G''_\Delta(\sigma) \right) + \ldots$$

$$T_4(\sigma) = 2\pi \int_{\frac{d}{2}+i\mathbb{R}} \frac{d\Delta}{2\pi i} \Big( \frac{\varrho^{\mathcal{P}}_{\hat{T},2}(\Delta)}{(d+1)^2} \left( (1-\sigma^2)^2 \mathcal{G}_2(\sigma) + \sigma(\sigma^2-1)\mathcal{G}_1(\sigma) + (\sigma^2 - d - 2)\mathcal{G}_0(\sigma) \right)$$
$$+ \varrho^{\mathcal{P}}_{\hat{T},0}(\Delta) \left( (d+\Delta\bar{\Delta})^2 G_\Delta(\sigma) + \sigma(1+2d+2\Delta\bar{\Delta})G'_\Delta(\sigma) + \sigma^2 G''_\Delta(\sigma) \right) \Big) + \ldots$$

$$T_3(\sigma) = -\frac{2\pi}{\sigma} \int_{\frac{d}{2}+i\mathbb{R}} \frac{d\Delta}{2\pi i} \left( \frac{\varrho^{\mathcal{P}}_{\hat{T},2}(\Delta)}{4} \left( 2\mathcal{G}_0(\sigma) + \sigma\mathcal{G}_1(\sigma) \right) + \varrho^{\mathcal{P}}_{\hat{T},0}(\Delta) \left( G''_\Delta(\sigma) + \sigma G'''_\Delta(\sigma) \right) \right) + \ldots$$

$$T_2(\sigma) = 2\pi \int_{\frac{d}{2}+i\mathbb{R}} \frac{d\Delta}{2\pi i} \Big( \frac{\varrho^{\mathcal{P}}_{\hat{T},2}(\Delta)}{(d+1)} \left( (\sigma^2-1)\mathcal{G}_2(\sigma) + \sigma\mathcal{G}_1(\sigma) + \mathcal{G}_0(\sigma) \right) \tag{C.12}$$
$$- \varrho^{\mathcal{P}}_{\hat{T},0}(\Delta) \left( (d+2+\Delta\bar{\Delta})G''_\Delta(\sigma) + \sigma G'''_\Delta(\sigma) \right) \Big) + \ldots$$

$$T_1(\sigma) = \frac{2\pi}{\sigma^2} \int_{\frac{d}{2}+i\mathbb{R}} \frac{d\Delta}{2\pi i} \Big( \varrho^{\mathcal{P}}_{\hat{T},2}(\Delta) \left( \mathcal{G}_0(\sigma) + \sigma\mathcal{G}_1(\sigma) + \sigma^2\mathcal{G}_2(\sigma) \right)$$
$$+ \varrho^{\mathcal{P}}_{\hat{T},0}(\Delta) \left( 2G''_\Delta(\sigma) + 4\sigma G'''_\Delta(\sigma) + \sigma^2 G''''_\Delta(\sigma) \right) \Big) + \ldots$$

where the functions $\mathcal{G}_m$ are defined in (B.16), primes are derivatives with respect to $\sigma$, and the dots stand for contributions from complementary and exceptional series. In particular, the complementary series contributions are exactly the same, with the only difference being the domain of integration. We checked these equations in the case of the free massive boson.

In two dimensions, we can find the analogous relations for the $\mathcal{T}_i$ functions by again extracting the coefficients of the tensor structures (C.11) from (3.8) and then using (2.18). We obtain

$$\mathcal{T}_1(\sigma) = (1-\sigma^2) \int_{\frac{1}{2}+i\mathbb{R}} \frac{d\Delta}{2\pi i} \frac{\varrho^{\mathcal{P}}_\Theta(\Delta)}{N(\Delta)} \left( (\Delta\bar{\Delta}+5)G''_\Delta(\sigma) + \sigma(5G'''_\Delta(\sigma) + \sigma G''''_\Delta(\sigma)) \right)$$
$$+ \frac{3}{4\pi} \varrho^{\mathcal{D}_2}_{\hat{T}} \frac{1+\sigma}{(1-\sigma)^3} + \text{complementary} \tag{C.13}$$

$$\mathcal{T}_2(\sigma) = (1-\sigma^2) \int_{\frac{1}{2}+i\mathbb{R}} \frac{d\Delta}{2\pi i} \frac{\varrho^{\mathcal{P}}_\Theta(\Delta)}{N(\Delta)} \left( (\Delta\bar{\Delta}+5)G''_\Delta(\sigma) + 3\sigma G'''_\Delta(\sigma) \right) + \frac{3}{4\pi} \varrho^{\mathcal{D}_2}_{\hat{T}} \frac{1+\sigma}{(1-\sigma)^2}$$
$$+ \text{complementary} \tag{C.14}$$

$$\mathcal{T}_3(\sigma) = \int_{\frac{1}{2}+i\mathbb{R}} \frac{d\Delta}{2\pi i} \frac{\varrho^{\mathcal{P}}_\Theta(\Delta)}{N(\Delta)} \Big( (\Delta\bar{\Delta}+1)^2 G_\Delta(\sigma) + (3+2\Delta\bar{\Delta})\sigma G'_\Delta(\sigma) + (2-\sigma^2)G''_\Delta(\sigma)$$
$$+ 2\sigma(1-\sigma^2)G'''_\Delta(\sigma) \Big) + \frac{3}{8\pi} \varrho^{\mathcal{D}_2}_{\hat{T}} \frac{1+2\sigma}{(1-\sigma)^2} + \text{complementary} \tag{C.15}$$

$$\mathcal{T}_4(\sigma) = \int_{\frac{1}{2}+i\mathbb{R}} \frac{d\Delta}{2\pi i} \frac{\varrho^{\mathcal{P}}_\Theta(\Delta)}{N(\Delta)} \Big( (\Delta\bar{\Delta}+1)^2 G_\Delta(\sigma) + (3+2\Delta\bar{\Delta})\sigma G'_\Delta(\sigma) + (2+\sigma^2)G''_\Delta(\sigma) \Big)$$
$$+ \frac{3}{8\pi} \varrho^{\mathcal{D}_2}_{\hat{T}} \frac{1}{(1-\sigma)^2} + \text{complementary}\,, \tag{C.16}$$

where here

$$N(\Delta) \equiv \frac{(\Delta + 1)^2 (\bar{\Delta} + 1)^2}{\pi} \,. \tag{C.17}$$

By plugging in $\sigma = -1$ and using that the $n$-th derivative of the scalar propagator at antipodal separation is

$$\partial_\sigma^n G_\Delta(\sigma) \Big|_{\sigma=-1} = \frac{\Gamma(\Delta + n)\Gamma(\bar{\Delta} + n)}{2^{2+n}\pi n!} \,, \tag{C.18}$$

we find that all the integrands for $\mathcal{T}_i(-1)$ go like $e^{i\pi\Delta} \varrho_\Theta^{\mathcal{P}}(\Delta)$ as $\Delta \to \frac{1}{2} + i\infty$. They thus converge if $\varrho_\Theta^{\mathcal{P}}(\Delta)$ does not grow exponentially in that same limit. This limit corresponds to the flat space limit, and in flat space spectral densities can only grow polynomially. We thus proved that the $\mathcal{T}_i$ functions are analytic around $\sigma = -1$. Moreover

$$\mathcal{T}_1(-1) = \mathcal{T}_2(-1) = 0 \,. \tag{C.19}$$

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
