# Peer review of "RG flows in de Sitter: c-functions and sum rules"

_SciPost Physics_

## Round 1 · Referee Report · Anonymous (Referee 1) · 2024-7-17

Strengths

1- Clear exposition in the main body 2- Technicalities in the appendices

Weaknesses

1- Example of free boson/Schwinger model: the function $c_1(R)$ does not satisfy the property $\lim_{R\to 0}c_1(R) = c^{UV}$ 2- Confusing structure of the appendices that does not reflect the main body

Report

The manuscript presents the study of RG of QFTs on $dS_2$ and $S^2$. In particular, it is focused on the definition of two functions $c_{1,2}$ of the radius $R$ of $dS_2 / S^2$ that interpolate between the central charges of the CFTs in the UV and IR at the extremes of the RG flow. Such extremes are formally achieved with the two limits $R\to 0$ (UV) and $R\to\infty$ (IR). Both functions are built from the stress-energy tensor.
The paper is well written, especially in the main body. All the technical details are proven and the mathematical structure consists of a good level of accuracy and rigor.
The main criticality is outlined in the following: in section 2 a general discussion is carried on regarding the construction and properties of the first $c$-function $c_1$. In particular, in equation (2.45) is stated that $\lim_{R\to 0}c_1(R) = c^{UV}$. Then, in section 4, three basic physical examples are studied: free scalar boson, free scalar fermion and Schwinger model. The latter is claimed to be equivalent to the first one. In these two examples, the author shows that $c_1(R) = 0, \ \forall R$ (see below eq. (4.5)). This is formally in disagreement with eq. (2.45), since $\lim_{R\to 0}c_1(R) =0\ne c^{UV}=1$. The physical explanation regarding the illness of the definition of the massless theory in de Sitter is convincing, but from a mathematical point of view, this is a counterexample with respect to the previous general statement.
The structure of the appendices is more confused with respect to the main body and it does not follow the same structure, e.g. section 4 is paired with appendix B, while appendix C contains the technical details of previous sections.
Overall, the study presented is a relevant contribution to the knowledge of QFTs on dS. The comprehensive analysis and rigorous methodology employed in the research underscore its relevance and potential impact on further studies in this domain.

Requested changes

1- On page 9, usually the notation for set subtraction is with a backslash rather than a slash, (\setminus in Latex), so it would be clearer to write $p\in\mathbb{N}\setminus{0}$. 2- Regarding the violation of $\lim_{R\to 0}c_1(R) = c^{UV}$ in the free boson case. I suggest that the author modify the general statement (2.45) including a technical hypothesis that excludes the mentioned examples, and show in section 4.1 that such hypothesis is violated. 3- The structure of the appendices is confusing, since it does not follow the structure of the main text. I suggest that the author put them in the same order. 4- Typo of page 28: change symmetricty $\leftrightarrow$ symmetricity.

Recommendation

Ask for minor revision

  • validity: good
  • significance: good
  • originality: good
  • clarity: good
  • formatting: good
  • grammar: good

Author:  Manuel Loparco  on 2024-07-25  [id 4652]

(in reply to Report 1 on 2024-07-17)

I am grateful for the referee's comments and will implement their suggestions. Let me address the main content-related point they made. I will implement the rest of the referee's more formal comments in the new version of the paper.

2- Regarding the violation of $\lim_{R\to0}c_1(R)=c^{\text{UV}}$ in the free boson case. I suggest that the author modify the general statement (2.45) including a technical hypothesis that excludes the mentioned examples, and show in section 4.1 that such hypothesis is violated.

This will be done in the new version of the paper. The technical hypothesis (that was stated but not emphasized enough) is that the two-point function of the trace of the stress tensor needs to vanish as $R\to0$ in order to have $\lim_{R\to0}c_1(R)=c^{\text{UV}}$. This hypothesis is physically motivated by the fact that $R\to0$ is like taking the mass scales $m_i\to0$, hence probing the CFT in the UV, where $\langle\Theta\Theta\rangle=0$. Interestingly, starting from the Lagrangian of the massless free scalar one obtains $\langle\Theta\Theta\rangle=0$ (just take a trace in (4.2) after setting $m=0$). Instead, starting from the massive theory and taking the limit gives $\lim_{R\to0}\langle\Theta\Theta\rangle=\frac{1}{8\pi^2}$, thus violating our assumption. Similarly, I will add a technical assumption for $\lim_{R\to0}c_2(R)=c^{\text{UV}}$. The requirement in that case becomes that $\lim_{R\to0}\text{Disc}[\langle\Theta\Theta\rangle]=0$, following from the position space sum rule. This hypothesis is still satisfied in the free boson case, which is why $\lim_{R\to0}c_2(R)=c^{\text{UV}}$ in that case.

I have discussed some more details related to this point in my response to the other referee.

---

## Round 1 · Referee Report · Edoardo Lauria (Referee 2) · 2024-7-22

Report

This work studies unitary QFTs on a two-dimensional Euclidean sphere and on de Sitter space. The main result is a proof of existence of two functions of the radius $R$ of the sphere (de Sitter), $c_1(R)$ and $c_2(R)$, which interpolate between the UV ($mR\rightarrow 0$) and the IR ($mR\rightarrow \infty$) central charges of the critical QFT.

The function $c_1(R)$ is an integrated two-point correlator of stress-tensor's trace, with a specific kernel. The function $c_2(R)$ is derived from the spectral decomposition of the stress-tensor two-point function, and it is proportional to the spectral weight of its traceless part in the $\Delta=2$ discrete series of unitary and irreducible representations.

As a further check of the main results of the paper, the author considers two examples: a free massive scalar and a free massive fermion. In such examples, $c_1(R)$ and $c_2(R)$ are found to be monotonic for intermediate values of $mR$. There is no claim that it should be the case for more general theories.

The scientific quality of this work is high: it represents an important step towards a better understanding of QFTs in dS (expectation 2). The paper also meets all the general acceptance criteria. I have a few minor requests (mostly about presentation), before I can recommend this paper for publication:

Requested changes

  1. Introduction:

1.1 - The author states that: "In this work, we focus on RG flows in reflection positive QFTs on a two-dimensional Euclidean sphere $S_2$, or equivalently unitary QFTs in two-dimensional de Sitter spacetime $dS_2$". I find the adverb "equivalently" slightly confusing here;

1.2 - The author states that: "In contrast to the F-theorem and its generalization, $c_1(R)$ and $c_2(R)$ are related to correlation functions of a local operator, namely the stress tensor." Why is this "in contrast" with the F-theorem and its generalization?

1.3 - At pag 3, the author states that the "radius of the sphere provides a valuable IR regulator". But this is not always true, as seen in the free scalar example. See also related comments below.

  1. Section 2:

2.1 - At the beginning of section 2, it would be beneficial for the reader to have a short paragraph to recall the logic of the derivation, which at the moment I find a bit scattered around; Moreover, it looks like the sphere and dS case are treated on equal footing, but then starting from section 2.3 only dS is mentioned, while in section 3 they appear both;

2.2 - In the derivation of eq. (2.25), are the conservation equations really $E_i =0$? If yes, the author can write something like "Call $E_i = 0$ [...] the three conservation equations". If not, then it is not clear why solving eq. (2.25) would help with solving eq. (2.22);

2.3 - Just below eq. (2.25): I am bit confused by the logic of the derivation. Are $\mathcal{T}$ and $\mathcal{T}'$ treated as independent functions? What are you solving for? Where do I find the solutions for $g_i(\sigma)$ and $q_i(\sigma)$?

2.4 - How and where IR divergence might invalidate the derivation of eq. (2.30)? A comment might be useful here, since $c_1(R)$ turns out to be ill defined for the free massive scalar, precisely for this reason;

2.5 - At the beginning of section 2.3, the author states that: "we are excluding the possibility that, in two dimensions, the discrete series of UIRs $(\Delta = p \in \mathbb{N}/{0})$ could contribute to the K-L decomposition of the trace of the stress tensor, since it is a scalar operator. " Is then eq. (2.35) only a conjecture? Does this assumption play a role in the derivation of $c_2(R)$? If this is the case, then it might be stated where appropriate.

  1. Section 3:

3.1 - Where was eq. (3.3) derived? 3.2 - Does the result in eq. (3.12) use the assumption that discrete series of UIRs $(\Delta = p \in \mathbb{N}/{0})$ does not contribute? (see my previous comment).

  1. Section 4:

Below eq. (4.5), the author finds for a free massive scalar that $c_1(R)=0$ for all $R$, in contradiction with the main claim that $c_1(R)$ should interpolate between UV and IR. It is argued that this is an IR issue due to a divergent zero mode for the free massless scalar. What is this divergence and how does it imply eq. (4.6)? More generally, it is not very clear when $c_1(R)$ is useful. Should it be regularized? Is there a way to make it working for the free scalar? This issue is related to my more generic requests, at points 1.3 and 2.4 above.

  1. (Optional)

Shorten a bit the abstract and remove the figure. My opinion is that this figure does not help understanding the abstract and moreover it might lead the reader to think that the work is just about free bosons/fermions.

Recommendation

Ask for minor revision

  • validity: high
  • significance: good
  • originality: good
  • clarity: good
  • formatting: -
  • grammar: -

Author:  Manuel Loparco  on 2024-07-24  [id 4651]

(in reply to Report 2 by Edoardo Lauria on 2024-07-22)

I am grateful for the referee's report and their detailed comments. I will implement the formal suggestions. Here I address the comments about content.

1.2 - The author states that: "In contrast to the F-theorem and its generalization, $c_1(R)$ and $c_2(R)$ are related to correlation functions of a local operator, namely the stress tensor." Why is this "in contrast" with the F-theorem and its generalization?

The statement was meant to emphasize that $F$ is a purely non-local quantity (the log of the partition function), while instead $c_1$ and $c_2$ are inherently related to local correlation functions of the stress tensor. I agree the phrasing should be changed.

2.1 [...] it looks like the sphere and dS case are treated on equal footing, but then starting from section 2.3 only dS is mentioned, while in section 3 they appear both;

It is indeed the case that the sphere and dS are treated on equal footing throughout the paper, and if only one is mentioned at any point it is only in order to reduce redundancy. I will make this more clear at the beginning of the section.

2.2 - In the derivation of eq. (2.25), are the conservation equations really $E_i=0$? [...]

The conservation equations are indeed $E_i=0$.

2.3 - Just below eq. (2.25): I am bit confused by the logic of the derivation. Are $\mathcal{T}$ and $\mathcal{T}'$ treated as independent functions? What are you solving for? Where do I find the solutions for $g_i(\sigma)$ and $q_i(\sigma)$?

We want equation (2.25) to be valid at any point $\sigma$ and for any value of $\mathcal{T}_i(\sigma)$ and $\mathcal{T}'_i(\sigma)$. They can thus be treated as independent functions. Since there are four $\mathcal{T}$ and four $\mathcal{T}'_i$, this gives 8 independent equations, given by the requirement that their coefficients be zero. The unknowns we are solving for are the 8 functions $r(\sigma)$, $g_1(\sigma),\ldots,g_4(\sigma)$ and $q_1(\sigma),\ldots q_3(\sigma)$. I will rephrase this part for more clarity and include the explicit solutions for the $g$ and $q$ functions.

2.4 - How and where IR divergence might invalidate the derivation of eq. (2.30)? A comment might be useful here, since $c_1(R)$ turns out to be ill defined for the free massive scalar, precisely for this reason;

Equation (2.30) is true in any case, even in the free massive boson flow. What fails in that case is equation (2.45). In the argument to derive (2.45), it is stated that taking the radius to zero is like taking all mass scales to zero in units of the radius. That corresponds to approaching the UV CFT, where one would expect in general that the two-point function of the trace of the stress tensor vanishes. Interestingly, if one computes the trace of the stress tensor directly in the massless theory, its two-point function does indeed vanish. Instead, taking the massless limit of the massive theory does not return zero. This is the origin of the contradiction which makes $c_1$ fail in this theory. I will add a more explicit discussion of this assumption and how it is violated by the free boson in dS. See my answer to the referee's question on section 4 for more details.

2.5 - At the beginning of section 2.3, the author states that: "we are excluding the possibility that, in two dimensions, the discrete series of UIRs ($\Delta=p\in\mathbb{N}/{0}$) could contribute to the K-L decomposition of the trace of the stress tensor, since it is a scalar operator. " Is then eq. (2.35) only a conjecture? Does this assumption play a role in the derivation of $c_2(R)$? If this is the case, then it might be stated where appropriate.

Let me be precise about the status of this statement. I will add this discussion to the next version of the paper.

The rigorous statement is that the KL decomposition of a scalar operator cannot include one isolated discrete series contribution. This is a statement that follows from microcausality and locality: the Green's functions that solve the Casimir equation for $\Delta=p\in\mathbb{N}/{0}$ either grow with the geodesic distance or have unphysical branch cuts when the two points are spacelike separated. What instead has not been ruled out is the possibility that a sum of discrete series contributions appear with their coefficients tuned in such a way that the unphysical branch cuts are canceled. We expect that there should be a proof that this is impossible, but at the moment this possibility has not been rigorously excluded. Everything I just wrote applies more generally to the case of the two-point function of an operator with $J$ indices. In that case, the Casimir equation can be solved without incurring in the mentioned issues if $J\geq p$, hence why the stress tensor can have a $\Delta=2$ contribution (more discussions on these points are presented in references [45] and [50] in the paper). This assumption does plays indeed a crucial role in translating the position space sum rules into spectral densities sum rules. No counterexample to this assumption is present in the literature.

3.1 - Where was eq. (3.3) derived?

Citations will be added.

3.2 - Does the result in eq. (3.12) use the assumption that discrete series of UIRs ($\Delta=p\in\mathbb{N}/{0}$) does not contribute? (see my previous comment).

To derive (3.12) the assumption is that $\Delta>2$ discrete series contributions cannot appear. That is because the stress tensor is an operator with $J=2$ indices. See my answer to the referee's question 2.5 for more details.

Section 4. - Below eq. (4.5), the author finds for a free massive scalar that $c_1(R)=0$ for all $R$, in contradiction with the main claim that $c_1(R)$ should interpolate between UV and IR. It is argued that this is an IR issue due to a divergent zero mode for the free massless scalar. What is this divergence and how does it imply eq. (4.6)? More generally, it is not very clear when $c_1(R)$ is useful. Should it be regularized? Is there a way to make it working for the free scalar? This issue is related to my more generic requests, at points 1.3 and 2.4 above.

One of the ways to see this IR divergence is by taking the limit $mR\to 0$ in the two-point function of the free scalar field $G_\Delta(\sigma)$ (see for example equation (3.3)). In this limit, it diverges as $(mR)^{-2}$. When computing the two-point function of the trace of the stress tensor $G_\Theta(\sigma)=(mR)^4\left(G_\Delta(\sigma)\right)^2$, this divergence is precisely canceled by the factor of $(mR)^4$, see equation (4.6) on the paper. The fact that this constant is not zero shows that there is a difference between taking the limit $mR\to0$ and directly studying the massless theory. Moreover, it violates one of the assumptions in deriving equation (2.45). Interestingly, the sum rule for $c_2(R)$ does not depend directly on the two-point function of the trace of the stress tensor, but rather on its discontinuity. If one computes its discontinuity in this theory, one gets

$$ \text{Disc}[G_\Theta(\sigma)]=\frac{i}{4}(mR)^4\text{csc}(\pi\Delta)\mathbf{F}\left(\Delta,1-\Delta,1,\frac{1-\sigma}{2}\right)\left(1+2\mathbf{F}\left(\Delta,1-\Delta,1,\frac{1+\sigma}{2}\right)\right)\,, $$
where $\mathbf{F}$ is the hypergeometric function and $\Delta=\frac{1}{2}\left(1-\sqrt{1-(2mR)^2}\right)$. Taking $mR\to0$, this discontinuity vanishes, ensuring that $c_2(R)$ does interpolate all the way to $c^{\text{UV}}$, even though $c_1(R)$ does not. In other words, even if in a certain theory $G_\Theta$ does not vanish in the UV, if $\text{Disc}G_\Theta$ vanishes then $c_2$ will interpolate all the way to $c^{\text{UV}}$. I will add this discussion in the next version of the paper.

Anonymous on 2024-07-26  [id 4660]

(in reply to Manuel Loparco on 2024-07-24 [id 4651])

While implementing these changes in the new version of the paper I noticed I understated my answer to 2.5. Let me fix that.

The solutions to the Casimir equation for $\Delta=p\in\mathbb{N}\setminus\{0\}$ which do not grow at infinity have branch points at spacelike separation. I said one can only rigorously exclude the appearance of one single contribution with fixed $p$. Actually, one can also exclude the appearance of a finite number of contributions with different $p$'s. Infact, each discrete series propagator is a Legendre Q function, which has the property that its discontinuity is a Legendre P function. This means that the discontinuities of the various contributions are orthogonal to each other and thus can never be canceled. The only possibility that is not yet rigorously excluded is that one has an infinite sum of discrete series terms. The sum and the discontinuity don't commute, and so in principle, in absence of a proof, the exclusion of this possibility is a conjecture.

---

## Editorial Decision

resubmitted